



# Physiological and climate controls on foliar mercury uptake by European tree species

Lena Wohlgemuth[1], Pasi Rautio[2], Bernd Ahrends[3], Alexander Russ[4], Lars Vesterdal[5], Peter Waldner[6], Volkmar Timmermann[7], Nadine Eickenscheidt[8], Alfred Fürst[9], Martin Greve[10], Peter Roskams[11], Anne Thimonier[6], Manuel Nicolas[12], Anna Kowalska[13], Morten Ingerslev[5], Päivi Merilä[14], Sue Benham[15], Carmen Iacoban[16], Günter Hoch[1], Christine Alewell[1], Martin Jiskra[1]

[1] Department of Environmental Sciences, University of Basel, Basel, CH-4056, Switzerland
[2] Natural Resources Institute Finland (Luke), Ounasjoentie 6, 96200 FI-Rovaniemi, Finland
[3] Department of Environmental Control, Northwest German Forest Research Institute (NW-FVA), Grätzelstr. 2, D-37079 Göttingen, Germany
[4] Landesbetrieb Forst Brandenburg, Alfred-Möller-Straße 1, D-16225 Eberswalde, Germany
[5] Department of Geosciences and Natural Resource Management, University of Copenhagen, Rolighedsvej 23, DK-1958 Frederiksberg C, Denmark
[6] Swiss Federal Institute for Forest, Snow and Landscape Research (WSL), Zürcherstrasse 111, CH-8903 Birmensdorf, Switzerland
[7] Norwegian Institute of Bioeconomy Research (NIBIO), NO-1431 Ås, Norway
[8] State Agency for Nature, Environment and Consumer Protection of North Rhine-Westphalia (LANUV), Leibnizstr. 10, D-45659 Recklinghausen, Germany
[9] Department of Forest Protection, Austrian Federal Research Centre for Forests, Vienna, A-1130, Austria
[10] Landesforsten Rheinland-Pfalz, Hauptstr. 16, D-67705 Trippstadt, Germany
[11] Research Institute for Nature and Forest (INBO), Gaverstraat 4, B-9500 Geraardsbergen, Belgium
[12] Office National des Forêts (ONF), Département Recherche-Développement-Innovation, F-77300 Fontainebleau, France
[13] Forest Research Institute, Sekocin Stary, Braci Lesnej 3, PL-05-090 Raszyn, Poland
[14] Natural Resources Institute Finland (Luke), Paavo Havaksentie 3, FI-90570 Oulu, Finland
[15] Forest Research, Alice Holt Lodge, Farnham Surrey, GU51 3QE, United Kingdom
[16] Department of Ecology, "Marin Dracea" National Institute for Research and Development in Forestry, Campulung Moldovenesc Station, 73 bis Calea Bucovinei, RO-725100 Camulung Moldovenesc, Romania

*Correspondence to*: Lena Wohlgemuth (lena.wohlgemuth@unibas.ch), Martin Jiskra (martin.jiskra@unibas.ch)





**Abstract.**

Despite the importance of vegetation uptake of atmospheric gaseous elemental mercury (Hg(0)) within the global Hg cycle, little knowledge exists on the physiological, climatic and geographic factors controlling stomatal uptake of atmospheric Hg(0) by tree foliage. We investigate controls on foliar stomatal Hg(0) uptake by combining Hg measurements of 3,569 foliage

samples across Europe with data on tree species traits and environmental conditions. To account for foliar Hg accumulation over time, we normalized foliar Hg concentration over the foliar life period from the simulated start of the growing season to sample harvest.

The most relevant parameter impacting daily foliar stomatal Hg uptake was tree functional group (deciduous versus coniferous trees). On average, we measured 3.2 times higher daily foliar stomatal Hg uptake rates in deciduous leaves than in coniferous

needles of the same age. Across tree species, for foliage of beech and fir, and at two out of three forest plots with more than 20 samples, we found a significant ($p < 0.001$) increase in foliar Hg values with respective leaf nitrogen concentrations. We therefore suggest, that foliar stomatal Hg uptake is controlled by tree functional traits with uptake rates increasing from low to high nutrient content representing low to high physiological activity. For pine and spruce needles, we detected a significant linear decrease of daily foliar stomatal Hg uptake with the proportion of time, during which vapor pressure deficit (VPD)

exceeded the species-specific threshold values of 1.2 kPa and 3 kPa, respectively.  The proportion of time within the growing season, during which surface soil water content (ERA5-Land) in the region of forest plots was low correlated negatively with corresponding foliar Hg uptake rates of beech and pine. These findings suggest that stomatal uptake of atmospheric Hg(0) is inhibited under high VPD conditions and/or low soil water content due the regulation of stomatal conductance to reduce water loss under dry conditions. We therefore propose, that foliar Hg measurements bear the potential to serve as proxy for stomatal

conductance. Other parameters associated with forest sampling sites (latitude and altitude), sampled trees (average age and diameter at breast height) or regional satellite observation-based transpiration product (GLEAM) did not significantly correlate with daily foliar Hg uptake rates. We conclude that tree physiological activity and stomatal response to VPD and soil water content should be implemented in a stomatal Hg model, to assess future Hg cycling under different anthropogenic emission scenarios and global warming.


# 1 Introduction

Mercury (Hg) is a toxic pollutant that is emitted by anthropogenic and geogenic activities into the atmosphere, where it can be transported over large distances and is eventually transferred to terrestrial and ocean surfaces by dry or wet deposition (Bishop et al., 2020). From a public health perspective, transfer rates of Hg to aquatic ecosystems are particularly relevant within this

cycle, since Hg bioaccumulation in fish for consumption represents the most important Hg exposition pathway to many communities internationally (UN Environment, 2019). In order to constrain future Hg levels in edible fish and to assess how Hg exposure responds to curbed anthropogenic Hg emissions under the policies implemented by the 2017 UN Minamata





convention on mercury, it is important to understand and quantify all major net deposition fluxes within the global Hg cycle. Wet deposition occurs when water-soluble oxidized Hg(II) is washed out from the atmosphere with rainwater (Driscoll et al., 2013; Sprovieri et al., 2017) or by cloud water interception (Weiss-Penzias et al., 2012). In a dry deposition process, gaseous elemental Hg(0) and Hg(II) directly bind to surfaces (Bishop et al., 2020) or Hg(0) is taken up by plants (Zhou et al., 2021). For more than two decades, vegetation has been recognized as important vector for Hg(0) dry deposition within the terrestrial Hg cycle (Rea et al., 1996, 2002; Grigal, 2003). Based on this seminal work, researchers have since highlighted that vegetation impacts Hg levels of all other environmental compartments within the active Hg cycle (AMAP and UNEP, 2019; Bishop et al., 2020; Zhou et al., 2021). Vegetation uptake of Hg(0) governs the seasonality of atmospheric Hg(0) in the Northern Hemisphere with concentration minima in summer at the end of the growing season (Jiskra et al., 2018). Thus, vegetation has been suggested to operate like a global Hg pump (Obrist, 2007; Jiskra et al., 2018). Atmospheric Hg(0) taken up by vegetation is oxidized to Hg(II) within the plant tissue (Manceau et al., 2018) and transferred to soils via litterfall (Iverfeldt, 1991; Schwesig and Matzner, 2000; Rea et al., 2001; Graydon et al., 2008; Risch et al., 2012; Jiskra et al., 2015; Wright et al., 2016; Wang et al., 2016; Risch et al., 2017). Moreover, in forests, Hg deposition to the ground may occur by wash-off of Hg(0) from plant surfaces via throughfall and by Hg(0) uptake into woody tissues, lichen, mosses and soil litter (Wang et al., 2020; Obrist et al., 2021). Mercury sequestered by forest ecosystems accumulates in soil and may subsequently be transported from watersheds to streams, rivers and the ocean, where it can bioaccumulate in fish (Drenner et al., 2013; Jiskra et al., 2017; Sonke et al., 2018).

Concerning the mechanism of Hg accumulation in foliage, there are multiple lines of evidence that leaf stomata control the foliar Hg(0) uptake flux to terrestrial ecosystems: (i) Hg concentrations were found to be higher in internal foliar tissues than on leaf surfaces (Laacouri et al., 2013); (ii) experiments revealed that isotopic Hg tracers are transferred from the air to the leaf interior (Rutter et al., 2011); (iii) foliar Hg concentrations are associated with leaf stomatal density and net photosynthesis (Laacouri et al., 2013; Teixeira et al., 2018); (iv) the isotopic composition of foliage is discriminated in heavy isotopes compared to atmospheric Hg(0) (Demers et al., 2013; Enrico et al., 2016; Yu et al., 2016; Jiskra et al., 2019); (v) temporal and vertical variations of net foliar Hg(0) uptake fluxes in trees agree with the mechanism of stomatal Hg(0) uptake (Wohlgemuth et al., 2020). While there is increasing consensus that vegetation uptake of atmospheric Hg(0) occurs via the stomatal pathway, there remain research gaps regarding parameters regulating this stomatal Hg(0) uptake (Zhou et al., 2021). Consequently, the Hg(0) dry deposition flux to terrestrial surfaces in Hg Earth System Models is generally parametrized by static inferential or resistance-in-series approaches (Travnikov et al., 2017). Ecosystem processes, including canopy gas exchange, are sensitive to climate conditions (Running and Coughlan, 1988) and vary between different plant species (Reich et al., 2003). Trees control leaf diffusive gas fluxes through their stomata in order to optimize the diffusive influx of carbon dioxide for photosynthesis, while averting excessive loss of water vapor to the atmosphere (Körner, 2013). The balanced regulation of stomata allows trees to dynamically adjust their metabolism to climatic conditions (temperature, atmospheric humidity, vapor pressure deficit, solar radiation) and site-specific limitations (soil moisture, nutrient availability) under the constraints of tree-specific prerequisites (leaf structure, leaf life span, water use efficiency).



In this study we aim to improve the process understanding of the stomatal Hg(0) uptake with the long-term goal to advance the parameterization of the foliar Hg(0) uptake in Hg Earth System Models. The objectives of the study were: (i) to investigate how foliar Hg(0) uptake depends on the physiological traits of tree species, and (ii) to study how stomatal response of trees to climate conditions control foliar Hg(0) uptake. We address these objectives by analyzing a large dataset of foliar Hg uptake rates, tree functional traits and climate conditions across natural gradients in European forests covering various tree species and climate conditions.

## 2 Material and Methods

### 2.1 Foliage sampling and data set description

The final dataset for this study comprises Hg concentrations of 3,515 foliage samples from 2015 and 2017, of which 2,071 samples were provided by 17 participating countries of the UNECE International Co-operative Programme on Assessment and Monitoring of Air Pollution Effects on Forests (ICP Forests). The samples include sun exposed leaves and needles from the upper third of the tree canopy of typically 5 trees (Austrian Bio-Indicator Grid: 2 trees) of the main species on the plot taken during full development in summer (deciduous species) or at the onset of dormant season in autumn (evergreen species) using harmonized national methods according to the ICP Forests Manual (Rautio et al. 2016) as described e.g. in (Jonard et al., 2015). Around 10% of samples were taken during winter needle sampling campaigns (December until March). Sample preparation procedure typically includes separation of needle age classes, drying, milling and chemical analyses for macronutrients and further drying of a subsample at 105°C to constant weight for the determination of dry weight. The participating ICP Forests countries harvested and carried out these preprocessing steps and collected the associated metadata. Hg measurements of samples from ICP Forests Level II plots were performed at the University of Basel. Additional foliar Hg concentration data of 1444 samples from the Austrian Bio-Indicator Grid organized by the Austrian Federal Research Center for Forests (German acronym BFW) (BFW, 2016) were included in the analysis. The combined dataset consists of 3,569 foliage samples encompassing 23 species of coniferous and deciduous trees (Table S1). The most frequent (number of samples > 100) species within the dataset are Norway spruce (*Picea abies*; n = 2045), Scots pine (*Pinus sylvestris*; n = 397), European beech (*Fagus sylvatica*; n = 365), silver fir (*Abies alba*; n = 162), sessile oak (*Quercus petraea*; n = 133), Austrian pine (*Pinus nigra*; n = 122) and common oak (*Quercus robur*; n = 102). We pooled individual tree species into groups of tree species genera (e.g. beech, oak, pine, spruce), see Table S1. Coniferous samples consist of needles of different age classes: most of the needle samples (n = 1911) flushed in the sampling season (current season; $y_0$), 602 samples are one-year old ($y_1$), 120 samples are two-year old ($y_2$), 125 are three-year old ($y_3$), 22 samples are four-year old ($y_4$), 60 samples are five-year old ($y_5$) and 3 samples are six-year old ($y_6$) needles. Foliage samples originate from 992 European sites: 237 sites are ICP Forests Level II forest monitoring plots, 737 locations are sampling sites of the Austrian Bio-Indicator grid and the remaining (18) are sites which are not classified within the ICP Forests program. See Figure 1 for a geographic overview of foliage sampling sites from the sampling year 2017.

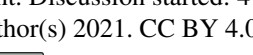



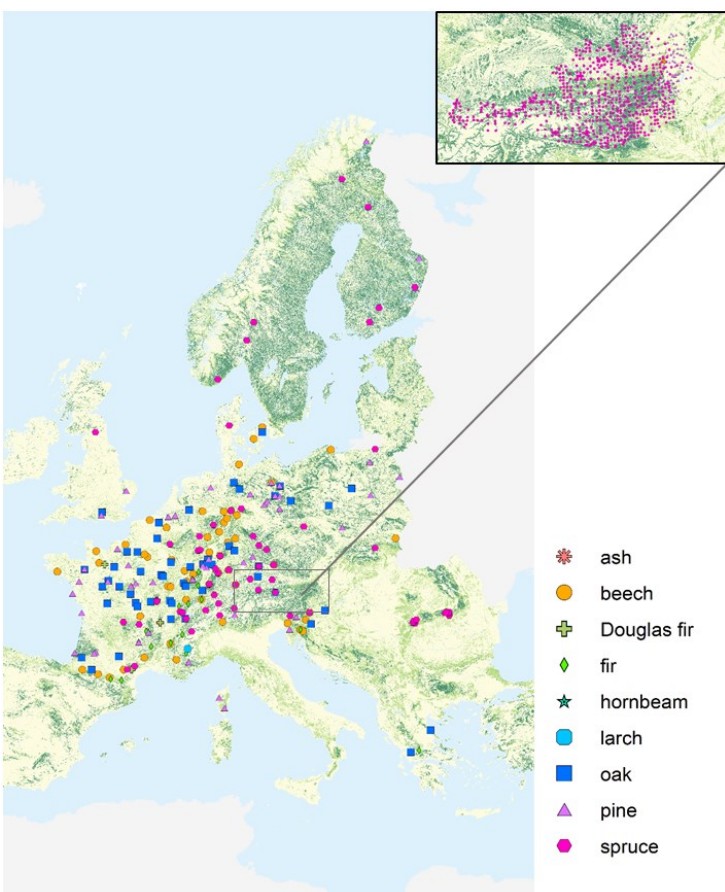

**Figure 1.** Overview of forest plots, at which Hg foliage samples were harvested from different tree species groups during the sampling year 2017. At around 12% of plots in 2017, foliage from more than one tree species group was sampled. Geographic distribution of sampling sites in 2015 is similar, except there were no samples from the ICP Forests countries Brandenburg (Germany), Baden-Wuerttemberg (Germany) and Poland and there were samples from five additional plots in North Rhine-Westphalia (Germany), see Fig. S1. The enlarged map view at the top right depicts sampling locations of the Bio-Indicator Grid in Austria in 2015 and 2017. Use of base
map authorized under European Commission reuse policy (EU, 2011).

We assembled the foliar Hg concentration dataset including the following metadata: sampling date, needle age class, leaf mass per area (LMA; 19% of samples), drying temperature, leaf nitrogen (N) and organic carbon ($C_{org}$) concentration. Foliage concentrations of N and $C_{org}$ were measured in laboratories in respective ICP Forests countries following strict QA procedures.

The tolerable quality limit for N concentration measurements is ± 15%  (for N concentration > 5 mg g$^{-1}$) of the mean interlaboratory N concentration in foliar reference material distributed for ICP Forests laboratory comparison tests (Rautio et al., 2016).

The used measurements and observation from Level II forest plots additionally included the beginning of the growing season for the sampling years 2015 and 2017 (where available) (Vilhar et al., 2013), main tree species on the plot, mean age of trees

on the plot (estimated during system installment), basal area and trees per hectare on the plot (Dobbertin and Neumann, 2016),



soil texture of the upper soil layer (mineral soil between 0 – 5 cm or 0 – 10 cm from the survey years 2003 - 2019) (Fleck et al., 2016; Cools and De Vos, 2020) as well as altitude and geographic coordinates. At the tree level, meta data consist of tree species, tree number and diameter at breast height (Dobbertin and Neumann, 2016). Meteorological in-situ measurements of hourly temperature and relative humidity  (Raspe et al., 2013) were available for 82 forest Level II plots for both 2015 and

155    2017.

Furthermore, we amended the dataset with satellite-based values of transpiration from the Global Land Evaporation Amsterdam Model (GLEAM) (Miralles et al., 2011; Martens et al., 2017), of hourly soil water (layer 1, 0 – 7 cm) and surface air temperature (2 m height) from ERA5-Land (Muñoz Sabater, 2019) for the respective regions of every forest plot. GLEAM (v. 3.3a) data were available at daily resolution and on a 0.25° latitude-longitude regular grid. ERA5-Land values were

available at hourly resolution and on a 0.1° latitude-longitude regular grid.  For each forest plot, we calculated average daily GLEAM (v.3.3a) transpiration within the life period of foliage samples, from the beginning of the growing season to harvest. Similarly, from ERA5-Land values, we calculated the average 2 m air temperature within respective sample life periods. We detected outliers of time-normalized foliar Hg concentrations (see Sec. 2.3) within each tree species and needle age class by applying the modified Z score method according to (Iglewicz and Hoaglin, 1993) using an absolute threshold value of 3.5,

above which a modified Z score value was considered an outlier. As a result, 3.2% of values within the dataset were removed as outliers.

## 2.2 Correction of foliar Hg concentrations for drying reference temperature

Drying and grinding of foliage samples were carried out by ICP Forests laboratories and BFW. All foliar concentration values (Hg, N and $C_{org}$) within the dataset are normalized to dry weight for a sample drying temperature of 105 °C in order to make

values internally consistent. The actual drying temperature differed between foliage samples (40 °C – 80 °C). In order to correct actual drying temperature, the laboratories determined the drying factor to correct for water content of each sample by drying an aliquot of foliage sample at actual drying temperature and subsequently at 105 °C. The drying factors were available for 62 % of samples within the dataset. For the rest of the samples an average drying factors per tree species and needle age class was applied for drying temperature correction. Smallest average drying factor was $1.03 \pm 0.003$ (mean ± sd) for one-year

old ($y_1$) *Pinus pinaster* needles and biggest average drying factor was $1.07 \pm 0.02$ (mean ± sd) for *Quercus robur* leaves. Previous studies did not detect Hg losses with drying temperature in foliage (Wohlgemuth et al., 2020), wood (Yang et al., 2017) or moss (Lodenius et al., 2003).

## 2.3 Foliage Hg analysis

Total Hg concentrations in foliage samples from ICP Forests Level II plots were measured at the University of Basel using a

direct mercury analyzer (Milestone DMA-80, Heerbrugg, Switzerland). Standard operation procedure involved measuring a pre-sequence of four blanks (three empty sample holders and wheat flour) and three liquid primary reference standards (50 mg of 100 ng g⁻¹ NIST-3133 in 1% BrCl). If the three liquid primary reference standards were within 90% - 110% of expected



value, we corrected all measurement results of the respective sequence accordingly. Otherwise, we discarded the sequence and re-calibrated the instrument. Standard reference materials (SRM) (NIST-1515 apple leaves and spruce needle sample B from

the 19[th] ICP Forests needle/leaf interlaboratory comparison test) were measured in each sequence (4 SRM in sequence of 40 samples) and the sequence was discarded, if the measured SRM value was outside the certified uncertainty range (NIST-1515) or outside ± 10% of the expected concentration (ICP Forests spruce B). Absolute Hg content in wheat blanks within the sequence had to be < 0.3 ng. We successfully participated in the 21[st] (2018/2019), 22[nd] (2019/2020) and 23[rd] (2020/2021) ICP Forests needle/leaf interlaboratory comparison (ILC) test. Total Hg concentrations in foliage samples from the Austrian Bio-

Indicator Grid were measured using a Hg analyzer (Altec-AMA 254 HCS, Prague, Czech Republic). Standard operation procedure at BFW involved a pre-sequence of five blanks (empty nickel boats) and measurements of three samples of reference material (BCR-62 olive leaves or spruce needle samples from the 17[th] or 19[th] ICP Forests needle/leaf ILC test) after every 40[th] sample within a sequence. If the measurement results of the three reference samples were outside of 93% - 107% of expected value, a drift correction was performed. Final foliage Hg concentrations within the Bio-Indicator Grid represent average values

of at least two replicates.

## 2.4 Determination of the beginning of the growing season for calculating daily foliage Hg uptake rates

Mercury concentrations in leaves and needles have been demonstrated to increase linearly over the course of the growing season (Rea et al., 2002; Laacouri et al., 2013; Blackwell et al., 2014; Wohlgemuth et al., 2020). In this study, foliage samples within the data set were harvested at various points in time, making a direct comparison of measured Hg concentrations

unfeasible. We therefore calculated foliar Hg uptake rates (in ng Hg $g^{-1}_{d.w.}$ $d^{-1}$) by normalizing foliar Hg concentrations of samples to their respective life period in days from the beginning of the growing season (leaf flushing) to date of harvest. While dates of harvest were available for all samples, we determined the start of the growing season by combining available data sources with start-of-season modelling. These data sources comprise in-situ phenological observations, which were available for 15% of samples, and observations of leaf unfolding for coniferous tree species from the Pan European

Phenological database PEP725 (Templ et al., 2018). We assigned observations from PEP725 to the corresponding closest forest plot of the respective sampling year (2015 or 2017) by using the nearest neighbor function *matchpt* from the Biobase package in R (Huber et al., 2015), such that differences between PEP725 observation and forest plots did not exceed 3° in latitude, 30 m in altitude and closely matched longitude as possible. For details on the matching procedure and results see supplementary information, Sect. S2.1. To model the beginning of the growing season for deciduous trees, we utilized the leaf

area index (LAI) product by Copernicus Global Land Service based on PROBA-V satellite imagery at a resolution of 300 m and 10 days (Dierckx et al., 2014; Fuster et al., 2020) following a recommendation by Bórnez et al., 2020. For information on the model and quality assurance, refer to supplementary information, Sect. S2.2.





## 2.5 Evaluation of vapor pressure deficit (VPD) data

At 82 ICP Forests Level II plots (in total from both years 2015 and 2017), in-situ meteorological data at an hourly resolution were recorded in 2015 and 2017, for which we calculated hourly vapor pressure deficit (VPD) values at daytime (06:00 – 18:00 LT). The VPD represents the difference between the vapor pressure at saturation and the actual vapor pressure. We calculated saturated vapor pressure from average hourly air temperature using the August-Roche-Magnus formula (Yuan et al., 2019) and actual vapor pressure as the saturation vapor pressure multiplied by the average hourly relative humidity. These

VPD values were calculated exclusively for daytime hours (06:00-18:00 LT), because both Hg(0) and photosynthetic $CO_2$ uptake by trees are at maximum during the day (Obrist et al., 2021). From these daytime hourly VPD values at each forest plot, we calculated the proportion of hours within the daytime life period of the samples (from the beginning of the growing season to sampling day), during which VPD exceeded the four threshold values of 1.2 kPa, 1.6 kPa, 2 kPa and 3 kPa respectively. We chose these four VPD thresholds because they were reported in literature to incrementally induce leaf stomatal

closure of temperate forest trees, ranging from initial stomatal closure (around 0.8 kPa – 1 kPa (Körner, 2013)) to maximum stomatal closure (at around 3 kPa – 3.2 kPa (CLRTAP, 2017)). We calculated the average proportion of daily daytime exceedance hours of VPD > respective threshold value by normalizing total number of respective daytime VPD exceedance hours with total number of daytime hours during the corresponding sample life period.

## 2.6 Evaluation of ERA5-Land volumetric soil water contents

We calculated the time proportion within sample life periods, during which the volumetric soil water content in the region of the respective forest plots fell below a soil texture dependent threshold value ($PAW_{crit}$) where plants are expected to close their stomata due to limited water availability. To do this, we used the satellite-derived ERA5-Land data of hourly soil water in soil layer 1 (vertical resolution: 0 – 7 cm, horizontal resolution: 0.1° x 0.1°) (Muñoz Sabater, 2019) and data on soil texture of the respective forest plots, where available (Fleck et al., 2016). Field data from literature suggest, that plant stomata start to close,

once the plant available water (PAW) in the soil falls below a critical value ($PAW_{crit}$) (Domec et al., 2009; Grünhage et al., 2011, 2012). The soil PAW represents the difference between soil water at field capacity ($SW_{FC}$) and soil water at the permanent wilting point ($SW_{PWP}$). We calculated $PAW_{crit} = 0.5 \times PAW + SW_{PWP}$ following a recommendation by (Büker et al., 2012) and used $PAW_{crit}$ as the threshold value to calculate the proportion of hours within the respective sample life periods, during which soil water < $PAW_{crit}$. See Figure S4 for an exemplary time series of ERA5 soil water in the region of a forest plot

in France in 2015. Soil texture specific values for $SW_{FC}$ and $SW_{PWP}$ (Table S2) were obtained from (Saxton and Rawls, 2006).



## 3 Results and Discussion

### 3.1 Variation of foliar Hg concentrations with foliar life period

Average foliar Hg concentrations (mean ± sd) differed between tree species groups (see Table S1 for definition of tree species groups). Ash leaves exhibited highest Hg concentrations (32.2 ± 5.7 ng Hg $g^{-1}_{d.w.}$; n = 10), followed by beech leaves (25.5 ±

9.6 ng Hg $g^{-1}_{d.w.}$; n = 373), current-season Douglas fir needles (22.9 ± 6.7 ng Hg $g^{-1}_{d.w.}$; n = 27), hornbeam leaves (32.2 ± 5.7 ng Hg $g^{-1}_{d.w.}$; n = 10), oak leaves (20.8 ± 9.1 ng Hg $g^{-1}_{d.w.}$; n = 287), larch needles (13.4 ± 3.4 ng Hg $g^{-1}_{d.w.}$; n = 3), current-season spruce needles (11.8 ± 3.4 ng Hg $g^{-1}_{d.w.}$; n = 1509), current-season fir needles (11.4 ± 2.8 ng Hg $g^{-1}_{d.w.}$; n = 66) and current-season pine needles (11.0 ± 5.1 ng Hg $g^{-1}_{d.w.}$; n = 344). For all tree species sampled at more than 20 forest plots, we found significant (p < 0.05) positive trends of foliar Hg concentrations with respective sampling date within the growing season

(see Figure 2 exemplary for beech and oak and Fig. S5 for pine and spruce).

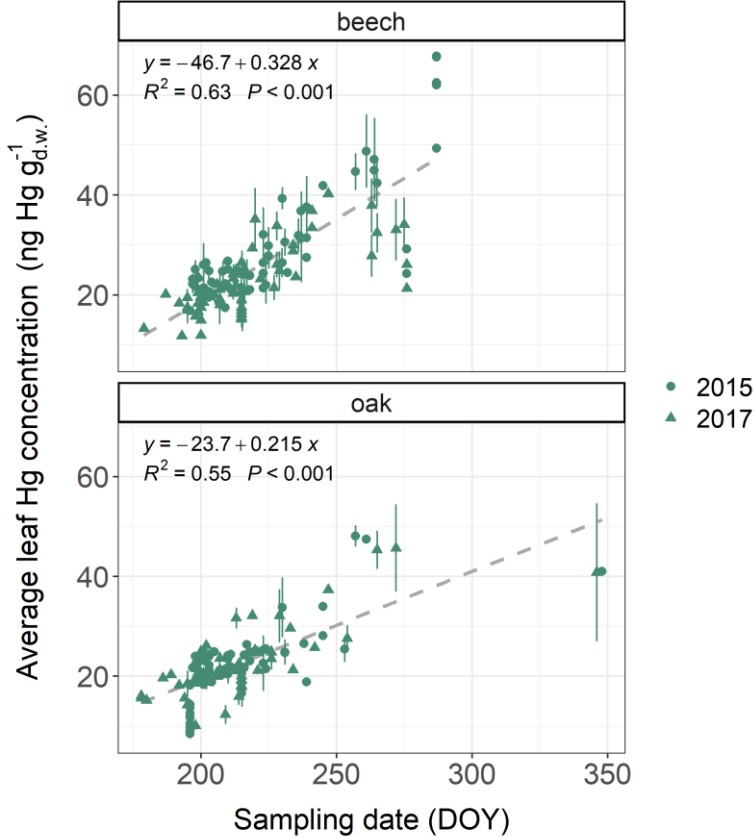

**Figure 2.** Average leaf Hg concentrations (ng Hg $g^{-1}_{d.w.}$) in beech and oak samples at multiple ICP Forests plots versus sampling dates (day of year; DOY) of respective samples. Sampling took place both in 2015 and 2017. Two plots of holm oak (*Quercus ilex*) are located in Greece and were sampled in December 2015 (DOY = 348) and December 2017 (DOY = 346). Error bars denote ± one standard deviation
between multiple samples at each forest plot.



Increasing foliar Hg concentrations with progressing sampling date are in line with previous observations demonstrating that at individual sites Hg concentrations increased linearly over the growing season (Rea et al., 2002; Laacouri et al., 2013; Wohlgemuth et al., 2020; Pleijel et al., 2021). To make Hg levels in foliage sampled at different times comparable, we calculated daily foliar Hg uptake rates by normalizing foliar Hg concentrations with the life period of samples. These daily

foliar Hg uptake rates represent average values over the life period. The average life period (mean ± sd) of samples was 104 d ± 30 d for beech, 104 d ± 24 d for oak, 159 d ± 12 d for pine and 148 d ± 14 d for spruce. At five percent of spruce plots sampling took place in winter (December until March). Spruce and pine trees have been found to reduce their physiological activity (transpiration, net photosynthesis) at low soil temperatures (< 8 – 10 °C), potentially impacting stomatal Hg(0) uptake in winter (Schwarz et al., 1997; Mellander et al., 2004). The average daily Hg uptake rates of current-season spruce needles

sampled during peak season (0.084 ng Hg g$^{-1}$$_{d.w.}$ day$^{-1}$) and sampled during winter (0.067 ng Hg g$^{-1}$$_{d.w.}$ day$^{-1}$) were significantly different (Welch two sample t-test; p = 0.015 at 95% confidence level). If spruce trees continue to accumulate Hg throughout the winter, Hg needle concentrations should be higher in winter samples than in samples harvested earlier during the growing season and Hg uptake rates per day should be comparable between winter and growing season samples. Thus, the difference of average daily Hg uptake between winter and growing season spruce needle samples indicates a decrease of Hg accumulation

in spruce needles during winter. However, the potential of needle Hg uptake in winter needles requires further investigation, e.g. by performing a full winter sampling at multiple forest plots. For this study, we shortened the calculated life period of spruce needles from winter sampling plots to 15$^{th}$ November (Rötzer and Chmielewski, 2001) to improve comparability of spruce needle Hg uptake rates within the dataset.

## 3.2 Variation of foliar Hg uptake rates with tree species groups

Median daily foliar Hg uptake rates (Fig. 3) in decreasing order are: ash (0.26 ng Hg g$^{-1}$$_{d.w.}$ d$^{-1}$), beech (0.25 ng Hg g$^{-1}$$_{d.w.}$ d$^{-1}$), oak (0.22 ng Hg g$^{-1}$$_{d.w.}$ d$^{-1}$), hornbeam (0.20 ng Hg g$^{-1}$$_{d.w.}$ d$^{-1}$), larch (0.14 ng Hg g$^{-1}$$_{d.w.}$ d$^{-1}$), current-season Douglas fir needles (0.13 ng Hg g$^{-1}$$_{d.w.}$ d$^{-1}$), current-season spruce needles (0.07 ng Hg g$^{-1}$$_{d.w.}$ d$^{-1}$), current-season fir needles (0.07 ng Hg g$^{-1}$$_{d.w.}$ d$^{-1}$) and current-season pine needles (0.05 ng Hg g$^{-1}$$_{d.w.}$ d$^{-1}$). The range of daily foliar Hg uptake of beech (0.12 – 0.42 ng Hg g$^{-1}$$_{d.w.}$ d$^{-1}$) is in agreement with the daily foliar Hg uptake rate of 0.35 ± 0.03 ng Hg g$^{-1}$ d$^{-1}$, that Bushey et al., 2008 had determined

in beech leaves growing in New York State in 2005. There are distinct differences in median daily Hg uptake rates between current-season foliage of tree species groups (Fig. 3). The median daily foliar Hg uptake rate of deciduous leaf samples is 0.23 ng Hg g$^{-1}$$_{d.w.}$ d$^{-1}$, a factor of 3.2 larger than the median daily foliar Hg uptake rate of current-season conifer needle values (0.07 ng Hg g$^{-1}$$_{d.w.}$ d$^{-1}$). The difference between deciduous and coniferous leaves in the European dataset is smaller than a previous observation from a mixed forest site in Switzerland in 2018, where Hg uptake rates of coniferous species were reported to be

5 times lower than those of deciduous trees (Wohlgemuth et al., 2020). Similarly, Zhou et al., 2021 reported higher foliar Hg concentrations in deciduous leaves (median: 28 ng Hg g$^{-1}$ from 341 remote sites) than in coniferous needles (median: 15 ng Hg g$^{-1}$ from 535 remote sites) based on global foliar Hg concentrations from literature, albeit Zhou et al., 2021 did not account





for life period of foliage samples and age of coniferous needles. Differences in daily foliar Hg uptake between tree species within one genus (e.g. *Quercus petraea* and *Quercus robur*) were negligible (see Fig. S6).

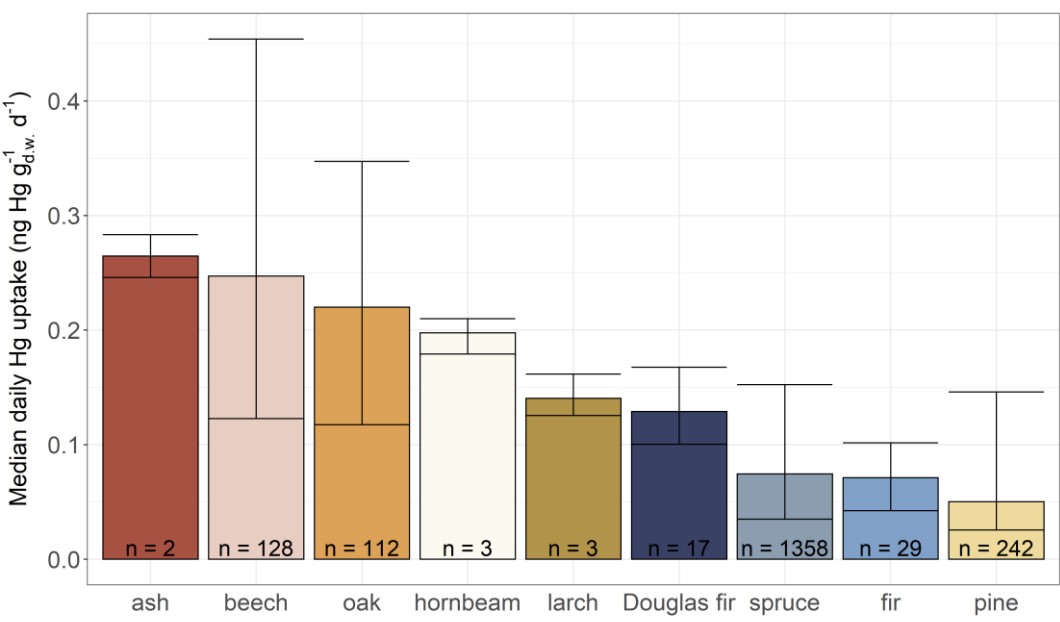


**Figure 3.** Median daily foliar Hg uptake (ng Hg g$^{-1}$$_{d.w.}$ day$^{-1}$) between different tree species groups (see Table S1 for definition of tree species groups) arranged from highest to lowest value. Error bars give the value range within each tree species group and n indicates the number of sites at which the respective tree species were sampled in both the years 2015 and 2017. Foliar samples of evergreen coniferous tree species (Douglas fir, spruce, fir and pine) consist of needles of the current season.

### 3.3 Foliar Hg uptake and sample-specific N concentration

Foliar N concentration serves as a surrogate for the maximum photosynthetic capacity of foliage (Reich et al., 1998) as the bulk amount of foliar N is contained in the photosynthetic systems like chlorophylls, thylokoid proteins and rubisco (Evans, 1989; Körner, 2013; Loomis, 1997). Furthermore, foliar N represents an indirect proxy for foliar maximum stomatal diffusive conductance for water vapor, independent of tree species (Körner et al., 1979; Bolton and Brown, 1980; Schulze et al., 1994;

Reich et al., 1999; Meziane and Shipley, 2001; Körner, 2013, 2018). Note, that for this analysis we solely compared N and Hg concentrations for foliage samples harvested within a period of the growing season, during which leaf N concentrations are relatively stable (July – August for broadleaves (Wilson et al., 2000; Mediavilla and Escudero, 2003) and Sept – March for conifer needles (Adams et al., 1987; Hatcher, 1990)). To assess the possibility of physiological factors controlling the large variation of foliar Hg(0) uptake between different tree species groups (Fig. 3), we compared average daily foliar Hg uptake

rates per tree species group with respective average foliar N concentrations. We found a positive linear correlation between foliar N concentration and Hg uptake rates with tree species group with high average foliar N exhibited higher daily foliar Hg uptake rates (1). This observation supports the notion, that the physiological activity of trees controls foliar Hg(0) uptake, thereby explaining the large variation among tree species groups (Wohlgemuth et al., 2020). We compared foliar Hg uptake





rates and leaf N concentrations with median stomatal conductance values for beech, oak, pine and spruce included in a global

leaf-level gas exchange database compiled by (Lin et al., 2015) (see description of database calculation in Sect. S2.4). Albeit stomatal conductance measurements for tree species of interest within the database (Lin et al., 2015) originated from one or only few sites (n = 1 – 5; Table 1), beech and oak exhibited higher median stomatal conductance values than spruce and pine, corresponding with higher daily Hg uptake rates and foliar N concentrations in beech and oak compared to spruce and pine. In other words, we observed a strong control of plant functional traits on foliar Hg(0) uptake with tree species of high

photosynthetic activity (high N concentration) and stomatal conductance exhibiting the highest foliar Hg(0) uptake rates.

**Table 1:** Mean ± standard deviation of daily Hg uptake and foliar N concentration per tree species group from a subset of foliage samples harvested during July – Aug (broadleaf samples) or Sept – March (coniferous needle samples). Values are ordered from highest to lowest mean daily Hg uptake. All values from evergreen tree species groups (Douglas fir, fir, pine, spruce) were evaluated in current-season needles.

Median stomatal conductance values (min – max) were calculated from a global database of leaf-level gas exchange parameters compiled by (Lin et al., 2015).

| Tree species group | daily Hg uptake (ng Hg g$^{-1}_{d.w.}$ d$^{-1}$) | | foliar N conc. (mg N g$^{-1}_{d.w.}$) | | n samples | median stomatal conductance (mol m$^{-2}$ s$^{-1}$) (Lin et al., 2015) | n sites (Lin et al., 2015) |
|---|---|---|---|---|---|---|---|
| beech | 0.25 ± | 0.05 | 23.1 ± | 2.9 | 312 | 0.10 (0.03 – 0.31) | 2 |
| oak | 0.20 ± | 0.05 | 25.1 ± | 2.8 | 252 | 0.15 (0.01 – 0.35) | 1 |
| hornbeam | 0.19 ± | 0.03 | 19.4 ± | 2.1 | 10 | | |
| Douglas fir | 0.13 ± | 0.02 | 17.0 ± | 3.5 | 26 | | |
| spruce | 0.08 ± | 0.02 | 12.9 ± | 1.7 | 1509 | 0.05 (0.01 – 0.16) | 1 |
| fir | 0.07 ± | 0.02 | 13.0 ± | 1.7 | 66 | | |
| pine | 0.06 ± | 0.02 | 14.4 ± | 3.0 | 355 | 0.06 (0.00 – 0.33) | 5 |

Within tree species groups, linear regression coefficients of daily Hg uptake and foliar N concentration were significant (p <

0.001) for beech ($R^2$ = 0.15; n = 312) and fir ($R^2$ = 0.27; n = 66). Corresponding statistical significance for hornbeam, oak, pine and spruce could not be evaluated, since the respective data used for the linear regression was heteroscedastic. Blackwell and Driscoll, 2015 found a significant relationship between foliar Hg concentration and foliar N% for yellow birch, sugar maple and American beech, but not for pine (red pine and white pine), red spruce or balsam fir. We examined whether unaccounted site-specific differences (e.g. soil N concentration) between forest plots potentially could have caused the

variability (low $R^2$) of daily Hg uptake versus foliar N concentration within tree species by individually analyzing foliar Hg concentration versus foliar N concentration at two oak and one beech forest plot, from which 20 or more foliage samples were





available. Linear regression coefficients of foliar Hg concentrations versus foliar N concentration were significant ($p < 0.001$) at two (oak and beech) of the three plots but not at the third plot ($p = 0.1$, oak) (see Fig. S7). This finding suggests that foliar N concentration represents an indicator of foliar Hg concentration at individual forest sites, as it does for foliar Hg uptake of

different tree species (Table 1). However, given the heterogeneity of nutrient availability between sites (Vesterdal et al., 2008) and the complexity of internal foliar allocation of N to different parts of the photosynthetic apparatus (Hikosaka, 2004), a generally valid correlation of foliar Hg uptake versus foliar N may not exist.

### 3.4 Foliar Hg uptake and Leaf Mass Area

Within the whole dataset, leaf mass per area (LMA; $g_{d.w.}$ $m^{-2}_{leaf}$) data were available in a subset of 338 foliage samples from

50 sites (from both 2015 and 2017). LMA is an important parameter in plant ecophysiology, because carbon gains of plants via photosynthetic activity and gas diffusion is optimized per unit of leaf area, as plants adapt their LMA, i.e. their foliage thickness and/or tissue density, to the availability of sunlight during growth (Ellsworth and Reich, 1993; Niinemets and Tenhunen, 1997; Rosati et al., 1999). This LMA adaptation of foliage to sunlight had been suggested to be more effective for optimizing photosynthetic capacity than within-leaf N partitioning of photosynthesizing biomass (Evans and Poorter, 2001).

Therefore, we analyzed the connection of foliar Hg uptake to LMA across tree species. Figure 44 shows average LMA values (mean ± sd) of the subset of samples where LMA was reported, resolved by tree species, along with respective average daily Hg uptake rates and associated foliar N concentrations.





**Figure 4.** Average daily Hg uptake rates (ng Hg $g^{-1}_{d.w.}$ $d^{-1}$), average foliar nitrogen concentrations (mg N $g^{-1}_{d.w.}$) and average LMA ($g_{d.w.}$ $m^{-2}_{leaf}$) determined in 338 foliage samples and resolved by tree species group and foliage type (leaf/needle). Error bars denote ± one standard deviation. Number of samples (n) differs between tree species: beech (n = 164), Douglas fir (n = 2), hornbeam (n = 9), oak (n = 106), pine (n = 24) and spruce (n = 33).

Current-season needle samples of coniferous tree species groups (Douglas fir, pine, spruce) exhibited higher median LMA values (308 $g_{d.w.}$ $m^{-2}_{leaf}$), lower median daily Hg uptake rates (0.10 ng Hg $g^{-1}_{d.w.}$ $d^{-1}$) and lower median foliar N concentrations (15.4 mg N $g^{-1}_{d.w.}$) compared to leaf samples of deciduous tree species groups (beech, oak, hornbeam) (Fig. 4). Wright et al. (2004) illustrated, that different evolutionary survival strategies of plant species are positioned along a single axis of foliage characteristics ranging from plant species with high photosynthetic capacity and respiration, high foliar N concentration, low LMA and short leaf life spans to plant species with the complete opposite attributes. Comparison of average daily foliar Hg



uptake, LMA and foliar N concentrations (Fig. 4) across tree species in this study suggest, that foliar Hg(0) uptake aligns along
this plant species economics spectrum, with deciduous leaves with high leaf N concentrations and thus high physiological
capacity (photosynthesis, respiration) taking up more Hg(0) than coniferous needles with low leaf N concentrations and
physiological capacity.

**3.5 Foliar Hg uptake and vapor pressure deficit (VPD)**

Trees regulate their transpiration rates in response to temporary changes of vapor pressure deficit (VPD) by controlling leaf
stomatal aperture (Franks and Farquhar, 1999; McAdam and Brodribb, 2015; Grossiord et al., 2020). When a critical VPD
threshold is exceeded, trees close their stomata to resist cavitation and excessive water loss in conditions of high atmospheric
evaporative forcing (i.e. high VPD) (Körner, 2013; Grossiord et al., 2020). This decrease in leaf stomatal conductivity in
response to high VPD suppresses stomatal uptake fluxes of gaseous pollutants like ozone (Emberson et al., 2000; Körner,
2013). We investigated, whether VPD impacts foliar uptake of gaseous Hg(0) by relating species-specific average daily foliar
Hg uptake rates to the proportion of daytime (06:00 – 18:00 LT) hours of an average day within the respective sample life
periods, during which hourly daytime VPD exceeded the threshold values of 1.2 kPa, 1.6 kPa, 2 kPa and 3 kPa respectively at
all forest plots with hourly meteorological data (n = 82 including both sampling years).

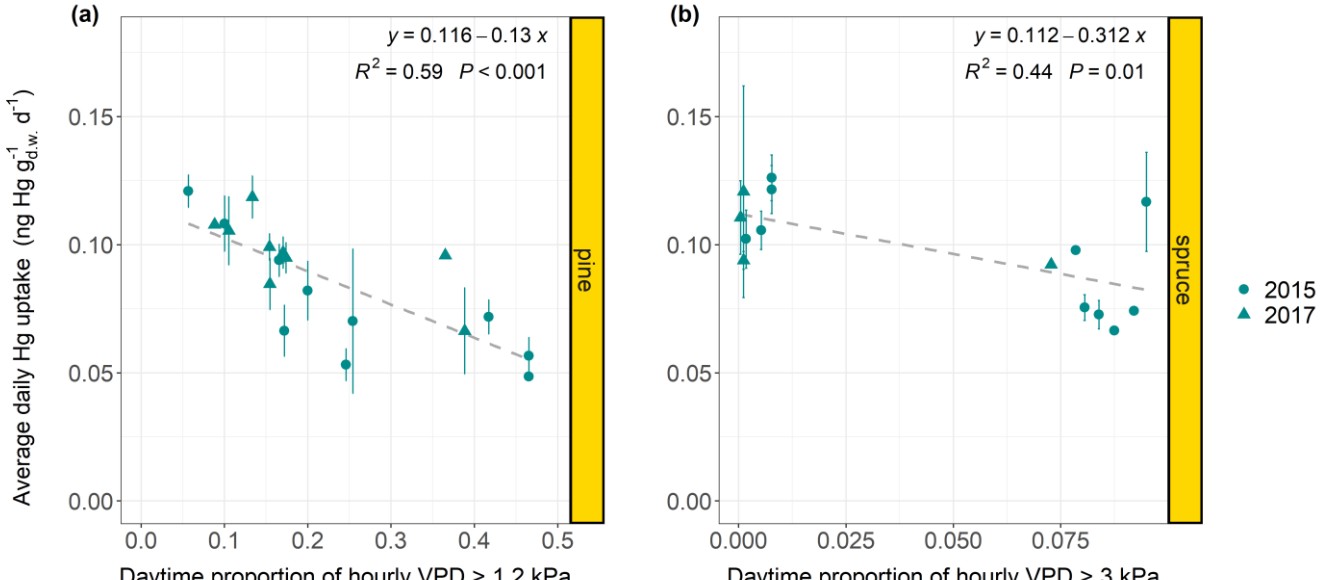

**Figure 5.** Average daily Hg uptake rates (ng Hg $g^{-1}_{d.w.}$ $d^{-1}$) of (a) current-season pine and (b) current-season spruce needles from multiple
forest plots (pine plots n = 19; spruce plots n = 14) versus the proportion of daytime hours (06:00 – 18:00 LT) within an average day of the
respective sample life periods, during which the hourly daytime vapor pressure deficit (VPD) exceeded a threshold value of (a) 1.2 kPa and
(b) 3 kPa. Data points originate from both sampling years 2015 and 2017. Error bars denote ± one standard deviation of daily needle Hg
uptake rates between multiple samples at each forest plot.

The linear regression coefficients of average daily Hg uptake versus daily proportion of daytime hours, during which VPD
exceeded a threshold value (1.2 kPa, 1.6 kPa, 2 kPa or 3 kPa) were significant (p < 0.01) for pine at all VPD threshold values





(Fig. 5a, Fig. S8), for spruce at a VPD threshold value of 3 kPa ($R^2 = 0.44$; $p = 0.01$; $n = 14$) (Fig. 5b) and not significant for beech and oak at any VPD threshold value (Fig. S9). Linear regression coefficients were negative for all species and VPD threshold values, i.e. by tendency, average daily foliar Hg uptake rates decreased with average proportion of daytime hours, during which VPD > respective threshold value (1.2 kPa, 1.6 kPa, 2 kPa or 3 kPa). We excluded Douglas fir, fir, hornbeam

and larch from the regression analysis due to a low number of forest plots ($n = 1 – 5$). The timing and degree of stomatal closure during dry conditions is tree species-specific (Zweifel et al., 2009; Tsuji et al., 2020). Tree species like pine and spruce are isohydric, i.e. they tend to respond to drought stress under high evaporative demand by closing their stomata earlier than anisohydric species like beech and oak (Martínez-Ferri et al., 2000; Zweifel et al., 2007; Carnicer et al., 2013; Coll et al., 2013; Cárcer et al., 2018). Among isohydric species, pine had been observed to reduce tree conductance and stomatal aperture during

the onset of dry conditions earlier and at a greater rate than spruce (Lagergren and Lindroth, 2002; Zweifel et al., 2009), while spruce was observed to keep stomata completely closed under drought stress (Zweifel et al., 2009). We hypothesize, that the significantly decreasing average foliar stomatal Hg uptake rates with daytime proportion of VPD > 1.2 kPa for pine (Fig. 5a) and of VPD > 3 kPa for spruce (Fig. 5b) possibly reflects the early physiological response of pine, and the high degree of stomatal closure under drought stress of spruce. Oak exhibits later stomatal closure at the onset of dry conditions and higher

stomatal aperture under drought stress than e.g. pine (Zweifel et al., 2007, 2009), which may be the reason why there was by tendency a negative, but not significant correlation coefficient of average foliar Hg uptake with daytime proportion of VPD > any threshold value for oak (Fig. S9).

### 3.6 Foliar Hg uptake and soil water content

Linear regression coefficients of average daily foliar Hg uptake rates at each forest plot versus proportion of hours within

sample life periods, during which ERA5-Land soil water fell below a soil texture specific threshold value ($PAW_{crit}$) (see Sect. 2.6), were negative for all tree species groups and significant for beech ($p = 0.036$) and pine ($p = 0.031$) (Fig. 6). The linear regression coefficient was not significant for oak ($p = 0.169$) and not available for spruce due to a low number of data points.





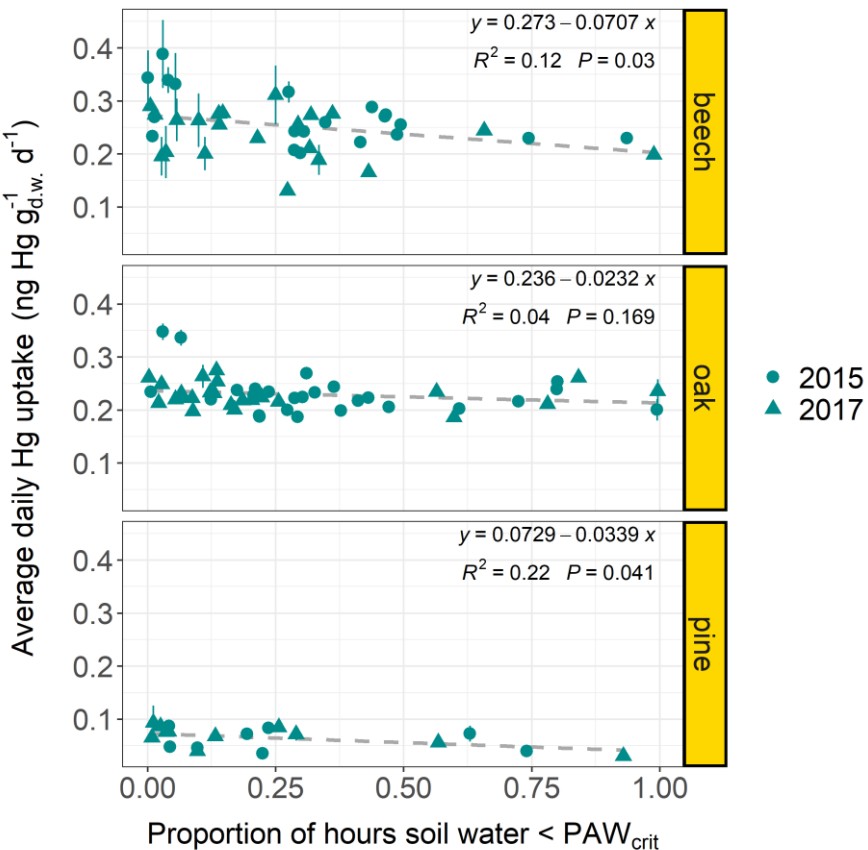

**Figure 6.** Average daily Hg uptake rates (ng Hg $g^{-1}_{d.w.}$ $d^{-1}$) of beech, oak and current-season pine foliage from multiple forest plots (beech plots n = 38; oak plots n = 45; pine plots n = 19) versus the proportion of hours within the respective sample life periods, during which the geographically associated hourly soil water from the ERA5-Land dataset (Muñoz Sabater, 2019) fell below a soil texture specific threshold value $PAW_{crit}$ (see Sect. 2.6). Data points originate from both sampling years 2015 and 2017. Error bars denote ± one standard deviation of daily needle Hg uptake rates between multiple samples at each forest plot.

Linear regression results (Fig. 6) indicate, that foliar Hg uptake rates decrease at forest plots, where plant available water in the upper soil layer (0 – 7 cm) falls below specific thresholds ($PAW_{crit}$) for a relatively long time period over the growing season. Studies on the atmosphere-plant transport of ozone have highlighted, that plant stomatal ozone uptake declines with increasing soil water deficit, because drought prompts stomatal closure (Panek and Goldstein, 2001; Simpson et al., 2003; Nunn et al., 2005). We hypothesize, that stomatal uptake of Hg(0) is impacted by soil conditions of low plant available water in a similar way to ozone. In future, in-situ soil matrix potential measurements should be used to better quantify the response rate of foliar Hg(0) uptake to soil water content in order to overcome the limitations of the coarse satellite-derived soil water measurements used here. We also suggest determining the possible influence of additional soil parameters like tree root depth, gravel or density, that we could not account for here.



### 3.7 Foliar Hg uptake and geographic and tree-specific parameters

We performed linear regressions of average daily foliar Hg uptake rates per forest plot and tree species group (beech, oak,
pine, spruce) versus geographic and tree-specific parameters. These parameters include altitude, latitude, average age of trees
on plot, average tree diameter at breast height, average daily GLEAM transpiration values and average ERA5-Land 2 m air
temperature over the course of the respective sample life periods (see Sect. 2.1). Out of the 54 sets of linear regression
parameters, the coefficient of average daily foliar Hg uptake rates of oak leaves was negative and significant versus latitude ($p$
= 0.004; $n$ = 112), was positive and significant versus daily average GLEAM transpiration values ($p$ = 0.007; $n$ = 110) and
versus average ERA5-Land 2 m air temperature with negligible $R^2$ = 0.07 respectively (see Fig. S10 – S12). For pine plots,
linear regression coefficient of average daily foliar Hg uptake rates versus daily average GLEAM transpiration was positive
and significant ($p$ = 0.046; $n$ = 76; $R^2$ = 0.05) (Fig. S13). The differences between 2015 and 2017 species-specific averages of
daily foliar Hg uptake rates from forest plots, at which foliage sampling took place during both sampling years, were small
compared to the standard deviation of daily foliar Hg uptake rates within each sampling year and species (see Table S3 for
average and standard deviation values). From the sampling year 2015 to the sampling year 2017 this difference was 0.04 ng
Hg $g^{-1}_{d.w.}$ $d^{-1}$ for beech, $2 \times 10^{-4}$ ng Hg $g^{-1}_{d.w.}$ $d^{-1}$ for oak, $8 \times 10^{-5}$ ng Hg $g^{-1}_{d.w.}$ $d^{-1}$ for pine (current-season needles) and $-3 \times 10^{-3}$ ng Hg $g^{-1}_{d.w.}$ $d^{-1}$ for spruce (current-season needles). We therefore suggest that differences in daily foliar Hg uptake rates
between the sampling years 2015 and 2017 are negligible. In agreement with previous studies (Ollerova et al., 2010; Hutnik
et al., 2014; Navrátil et al., 2019; Wohlgemuth et al., 2020; Pleijel et al., 2021), we found a clear trend of Hg concentrations
in differently aged needles with older needles exhibiting higher Hg concentrations (Fig. 7), demonstrating that Hg
accumulation continues in older needles.

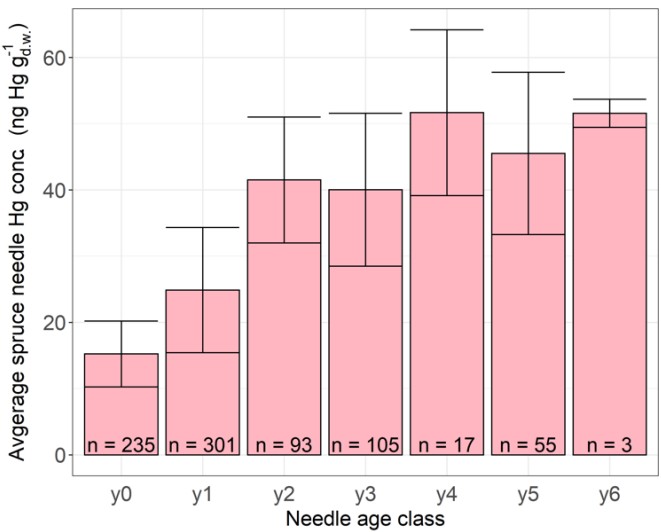

**Figure 7.** Average Hg concentrations (ng Hg $g_{d.w.}^{-1}$) in spruce needle samples of different ages. Needle age class $y_0$ corresponds to current-season needles developed in the year of sampling, $y_1$ corresponds to one-year old needles, $y_2$ corresponds to two-year old needles etc. Error
bars denote ± one standard deviation between multiple samples, n indicates number of samples.





### 3.8 Implications for Hg cycle modelling

Our findings suggest, that VPD impacts stomatal Hg(0) uptake by isohydric tree species due to stomatal closure during conditions of high VPD (Fig. 5). Similarly, elongated time periods of low soil water content within the growing season possibly result in a decrease of stomatal conductance to Hg(0) and thus in less foliar Hg(0) uptake by tree species such as beech and pine (Fig. 6). Other meteorological parameters such as temperature may also have an effect on stomatal closure and consequently stomatal Hg(0) uptake (Sect. 3.7). We therefore propose, to refine existing stomatal uptake models for the purpose of exploring the stomatal uptake flux of Hg(0) for common vegetation types across different global regions over the course of the growing season. For this, the sensitivity of foliar Hg uptake of different tree species to elevated VPD, low soil water content or temperature would have to be measured in laboratory experiments and the effect of tree species, VPD, soil water and point in time within the growing season could be implemented in a stomatal Hg deposition model in a multiplicative way. We propose, that the stomatal flux module of the DO3SE (Deposition of Ozone for Stomatal Exchange) model could serve as a prototype for a stomatal Hg deposition model, because DO3SE provides estimates of stomatal ozone deposition based on plant phenological and meteorological conditions (Emberson, L.D. et al., 2000). Projections from stomatal Hg models are particularly relevant for the evaluation of future global environmental Hg cycling, as the stomatal Hg(0) uptake flux exceeds direct Hg(II) wet deposition (Wohlgemuth et al., 2020; Obrist et al., 2021), and quantitatively represents the most relevant deposition pathways to land surfaces, driving the seasonality of Hg(0) in the atmosphere (Obrist, 2007; Jiskra et al., 2018). VPD is projected to increase with rising temperatures under global warming (Yuan et al., 2019; Grossiord et al., 2020), potentially causing a decrease in stomatal foliar Hg(0) uptake fluxes. A diminished global stomatal foliar Hg(0) uptake flux would result in higher Hg(0) concentrations in the atmosphere and higher Hg deposition fluxes to the ocean (Zhou et al., 2021).

### 4 Conclusion

We created a large European forest dataset for investigating the control of tree physiology and climatic conditions on foliar stomatal Hg(0) uptake. We observed, that foliar Hg concentrations were highly correlated with foliage sampling date (Fig. 2), confirming the notion, that foliage takes up Hg(0) over the entire life time. Consequently, it is necessary to calculate foliar Hg uptake rates by normalizing foliar Hg concentrations by the time period of Hg(0) accumulation to make foliar Hg values from different sites comparable. We found notable differences of daily foliar Hg uptake rates between tree functional groups (broadleaves versus coniferous needles), i.e. Hg uptake rates of broadleaves were higher compared to coniferous needles of the same age by a factor of 3.2 (Fig. 3). Across tree species and within beech, Douglas fir and fir, the linear regression coefficients of daily foliar Hg uptake rates versus foliar N concentration were significant (Sect. 3.3). Tree species groups with foliage of lower LMA exhibited higher daily rates of Hg uptake per dry weight of foliage (Sect. 3.4). We set these results within the context of stomatal foliar uptake of atmospheric Hg(0): Deciduous tree species like beech and oak, which exhibit functional traits of high physiological activity (photosynthesis, transpiration) over the time span of one growing season, as represented by high foliar N concentration and low LMA, retain a higher stomatal conductance for diffusive gas exchange.



Thus, beech and oak leaves accumulate more Hg per unit dry weight over the same time span relative to needles of coniferous tree species. In addition to tree species-specific metabolism, climatic conditions like current VPD or soil water content, which

impacts stomatal gas exchange, can affect foliar Hg uptake. For coniferous pine and spruce needles (current-season), we found a significant negative linear regression coefficient of daily Hg uptake rates versus the average daily proportion of hours within sample life period, during which atmospheric evaporative forcing was high (VPD > 1.2 kPa for pine and VPD > 3 kPa for spruce) (Fig. 5), suggesting, that a reduction of stomatal conductance during conditions of high VPD suppresses foliar Hg(0) uptake. In a similar line of argument, low surface soil water content lowers stomatal conductance and consequently foliar

stomatal Hg(0) uptake (Fig. 6). We therefore suggest, that foliar Hg measurements bear the potential to serve as a proxy for stomatal conductance, providing a time-integrated measure for stomatal aperture. We call for the implementation of a stomatal Hg(0) deposition model, that takes tree physiology and environmental conditions like VPD or soil water content into account, in order to make projections about this important Hg deposition flux under climate change. The diminution of the vegetation mercury pump in response to drought stress as a result of climate change could result in elevated Hg concentration in the ocean

and potentially in marine fish in future, a potential risk which warrants further quantitative studies.

***Acknowledgment.*** We thank Fabienne Bracher and Judith Kobler Waldis for assistance in foliage sample analysis. The evaluation was based on data that was collected by partners of the official UNECE ICP Forests Network (http://icp-

forests.net/contributors). We are grateful to all ICP Forests participants who supported the project through foliage sampling, nutrient analysis and cooperation in the logistics of this project. In this context, we particularly thank Martin Maier, Andrea Hölscher and their team from the Department of Soil and Environment at FVA Baden-Württemberg; Daniel Žlindra from the Slovenian Forestry Institute; Nils König from Northwest German Forest Research Institute (NW-FVA); Hans-Peter Dietrich and Stephan Raspe from the Bavarian State Institute of Forestry (LWF Bayern); Michael Tatzber from the Austrian Research

Centre for Forests (BFW); Arne Verstraeten and Luc De Geest from the Belgian Research Institute for Nature and Forest (INBO); Sébastien Macé from the French National Forest Office (ONF) and Panagiotis Michopoulos from the Forest Research Institute of Athens (FRIA). We are grateful to Samantha Wittke and Christian Körner for their helpful advice and support on leaf area indices and plant phenology. Special thanks go to Till Kirchner and Anne-Katrin Prescher from Thünen Institute for their assistance in accessing the ICP Forests Database.


***Data availability***. Foliar Hg concentrations, foliar Hg uptake rates and Hg related metadata are available for download at https://doi.org/10.5281/zenodo.5495179. Please note that coordinates (latitude, longitude) were round to minutes. R-scripts for data analysis and plots of this paper can be found at https://github.com/wohle/Hg_Forests.

ICP Forests proprietory data (N concentrations and forest plot attributes) fall under the publication policy of ICP Forests

(Annex II of (Seidling et al., 2017)) and can be accessed from the ICP Forests Database (http://icp-forests.net/page/data-requests) upon request from the Programme Co-ordinating Center (PCC) in Eberswalde, Germany. Foliar Hg concentration



values from the Austrian Bio-Indicator Grid can be obtained from BFW upon request (https://bfw.ac.at/rz/bfwcms.web?dok=3687). ERA5-Land data (Muñoz Sabater, 2019) was downloaded from the Copernicus Climate Change Service (C3S) Climate Data Store (https://cds.climate.copernicus.eu/#!/home). Data for the beginning of the

growing season of coniferous trees in 2015 and 2017 were provided by members of the PEP725 project (http://www.pep725.eu/). PROBA-V leaf area index values of 300 m resolution and GLEAM transpiration values from 2015 and 2017 were obtained from the VITO Product Distribution Portal (https://land.copernicus.eu/global/products/lai) and the GLEAM server (https://www.gleam.eu/) respectively.

*Financial support.* This research was funded by the Swiss National Science Foundation (SNSF), project number 174101. The participating countries from the UNECE ICP Forests Network funded the sampling using national funding; among funding agencies are the Natural Resources Institute Finland (Luke), Swiss Federal Institute for Forest, Snow and Landscape Research (WSL), Norwegian Institute of Bioeconomy Research (NIBIO), Norwegian Ministry of Agriculture and Food (LMD), Polish Forest Research Institute (IBL), Polish Ministry of Environment (grants No. 650412-650415), Northwest German Forest

Research Institute (NW-FVA), French National Forest Office (ONF), French Ministry of Agriculture, French Agency for Environment and Energy (ADEME), Slovenian Ministry of Agriculture, Forestry and Food (Public Forest Service, Assignment 1.3). Part of the data was co-financed by the European Commission. The Austrian Bio-Indicator Grid is operated by BFW with funding from the Austrian Ministry of Agriculture, Regions and Tourism.

*Author contributions*. LW managed the project, coordinated foliar Hg measurements, assembled the Hg dataset, performed the data analysis and wrote the manuscript. PR, BA, AR, LV, PW, VT, NE, MG, PR, AT, MN, AK, MI, PM, SB, DZ, CI supplied foliage samples and metadata and gave scientific input to the manuscript. AF contributed foliar Hg concentrations from the Austrian Bio-Indicator Grid and gave scientific input to the manuscript. GH and CA provided valuable scientific support. MJ designed and set up the SNF project (174101), provided valuable scientific support and contributed to manuscript

writing.

*Competing interests*. The authors declare that they have no conflict of interest.





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
