# Peer review of "Physiological and climate controls on foliar mercury uptake by European tree species"

_Biogeosciences, 2021_

## Author Comment (AC1)

**Response to Charles Driscoll**

The referee comments are in black, while the author comments are in **bold print** and blue.

The article "Physiological and climate controls on foliar mercury uptake by European tree species" by Wohlgemuth et al. [bg-2021-239] summarizes an important study which examines controls on foliar mercury concentrations across several European tree species. This article is important because the authors synthesize mercury patterns across a relatively large spatial area involving many observations of different tree species and interpret these patterns in the context of tree physiological traits and climate conditions which control stomatal conductance. While the study presents few new finings, the authors do a great job presenting, synthesizing and interpreting their findings in context of a rather broad literature. Also based on their observations, the authors make recommendations for global mercury models to improve simulation of the role of "global vegetation as a mercury pump" and allows for depiction of the effects of changing climate on this important process.

The manuscript is well written and well organized. I only have a few minor suggestions below. I recommend the publication of this paper pending minor revisions.

**Thank you a lot for your comments and the positive assessment of the manuscript.**

**Specific comments**

Line 80. …deposition to the **land surface** may …

**revised as suggested**

Lines 96, 176, 243, 304, 329 and 335. The wording should probably be "among" rather than "between".

**Thank you, we will change accordingly.**

Lines 152, 203, 215 and 414. *"in-situ"* should be in italics.

**We will set *in-situ* in italics throughout the manuscript.**

Line 179. … needles and **largest** average …

**revised as suggested**

Line 199. … at various **dates over the annual cycle**, making …

**We will change it accordingly.**

Figure 3. The authors might point out that the range of observations for species with a large number of observations is large compared to species with few observations and the reason for this variation.

A possible reason for the wider variation in daily Hg uptake rates for species of large sample numbers could the greater range of foliage N concentrations in these species (see figure below). In Sect. 3.3 we discuss a positive relationship between foliar N and Hg uptake. The pool of foliage sample sites of tree species like beech and spruce are more diverse in terms of nutrient availability than of tree species like larch and Douglas fir, which possibly results in a greater variation of daily Hg uptake rates of beech and spruce. We will add the sentence to the manuscript: "Variation in daily foliar Hg uptake rates within a given tree species group could result from differences in physiological activity or foliage structure (see Sect. 3.3 and 3.4)."

[Figure]

Line 345.  Average LMA values was …

In response to a comment by referee Håkan Pleijel we will give all values from Figure 4 including average LMA in a table in the Supplement. Therefore, the sentence of Line 345 will read: "Error! Reference source not found. shows average LMA values (mean ± sd) of the subset of samples where LMA was reported, resolved by tree species, along with respective average daily Hg uptake rates and associated foliar N concentrations (all values displayed in Fig. 4 are listed in Table S3; see Fig. S8 for density plots of datasets from Table 2 and Fig. 4)." We will give numbers of LMA values for broadleaves and coniferous needles later in this Sect. 3.4.

---

## Author Comment (AC2)

**Response to Frank Wania**

The referee comments are in black, while the author comments are in **bold print** and blue.

A well written paper on an important subject, based on a large new dataset that is analyzed thoroughly and rigorously. I particularly like the practical conclusions on how foliar uptake of mercury could be practically implemented in mercury fate models. I find little to criticise.

**Thank you for the positive feedback to this manuscript and your comments.**

Line 49: Delete comma before "during" and add "the water" before "vapor".

**revised as suggested**

Line 51: Delete comma before "during".

**revised as suggested**

Line 52: Delete "corresponding".

**revised as suggested**

Line 54: This sentence appears somewhat unmotivated in the abstract. Finding a proxy for stomatal conductance during an entire growing season was probably not among the original objectives of this study and is too hypothetical at this stage to merit inclusion in the abstract. It distracts from the main message. It also only merits a single sentence in the entire main body of the paper (line 480).

**We agree that our suggestion to use foliar Hg concentration as a proxy for stomatal conductance is not discussed enough in the manuscript to merit a mention in the abstract. We will therefore delete the sentence in the abstract.**

Line 65: use "exposure" instead of "exposition".

**revised as suggested**

Line 110 and line 122: 3515 foliage samples versus 3569 foliage samples. What is the reason for that discrepancy? Does either of these numbers include the outliers identified in line 165?

**Thank you for the careful reading. The number of Hg concentrations in the dataset, on which the analysis is based, is indeed 3,569. After revision of all numbers describing the dataset, we understood that there must have been a confusion when counting the total sample number brought together from the data sets of ICP Forests, non ICP Forests (but provided by ICP Forests countries, only few samples) and Austrian Bio-Indicator Grid, which were all used together in the analysis. The total number of values of this dataset after blank correction is 3,569 Hg concentrations measured in 2,129 samples provided by ICP Forests countries (from both ICP Forests and non ICP Forests plots;**

published at https://doi.org/10.5281/zenodo.5495179) and in 1,440 samples from the Austrian Bio-Indicator Grid. We will consistently change the numbers and equally check the numbers of subsets of samples per tree species and needle age class.

Line 137 to 138: Brandenburg and Baden-Wuerttemberg are not countries.

**We will change "ICP Forests countries" to "ICP Forests members".**

Line 200: The text prior to here makes reference to multiple needle year classes (Line 117/141, line 127-130). The sentence "by normalizing foliar Hg concentrations of samples to their respective life period in days from the beginning of the growing season (leaf flushing) to date of harvest" raises the question of how was this done for needles older than 1 season. (Judging from page 18 and figure 7, it appears that older needles were not subjected to the normalisation procedure and were used in an entirely separate analysis.)

**Initially, we calculated daily Hg uptake rates of needles by dividing Hg concentrations with the respective number of days during the current-season needle life period + 356 days * needle age in years (e.g. 0; 1 year; 2 years etc.). However, the problem with this approach is, that we expect needle Hg uptake rates to slow down in multiple year needles (Wohlgemuth et al., 2020), which makes an average needle Hg uptake rate over the whole life period of older needles non-representative for a single growing season. Furthermore, the needle Hg uptake rate in winter and early spring is still unclear. Due to these issues, and because we directly compare foliar Hg uptake rates of broadleaves and needles, we decided to use Hg data from older needles exclusively in a separate analysis presented in Sect. 3.7. To make this clear, we will amend the sentence: "We therefore calculated foliar Hg uptake rates of underline{current-season} samples by normalizing foliar Hg concentrations to their respective life period…"**

Line 216: Make it clear that you refer to water vapour here. This is advisable as there could in principle also be a mercury vapor pressure deficit.

**We will explicitly specify water vapor pressure.**

Line 250: Use "example" instead of "exemplary"

**revised as suggested**

Figure S5: Maybe state that current-season needles are displayed.

**The caption of Fig. S5 includes: "All samples represent current-season values."**

Section 3.2 The comparison of foliar uptake rates across space suggests that variations in GEM concentrations in the atmosphere (in space and time) are deemed not to be important. That is likely correct, but should still be stated explicitly.

This is also important as some of the parameters explored later (soil dryness, VPD) could have a geographic component. You have to exclude the possibility that some of the observed relationships with these

parameters are not artefact caused by a correlation of the parameters with atmospheric GEM concentrations (e.g. lower atmospheric GEM levels in the more water-stressed southern parts of Europe).

**We agree that variations in GEM concentrations between sites have to be discussed, since it could indeed impact the results of the analysis. Unfortunately, we do not have GEM data available from any of the sampled forest sites to normalize foliar Hg uptake rates. In order to get a better understanding of GEM variation in Europe, we downloaded accessible EMEP Hg air data from terrestrial European stations between May 1st and Sept. 30th 2015 and 2017 from the EBAS database (http://ebas-data.nilu.no/Default.aspx). Like this, we obtained 11,900 Hg(0) values from 6 background sites (see figure below). We excluded available values from one site (Iskrba) in 2015 because air Hg values were abnormally low ($0.41 \pm 0.13$ ng m$^{-3}$; mean $\pm$ sd).**

[Figure]

**Air Hg from EMEP measured at 6 different sites during May – Sept. 2015 and/or 2017.**

**We calculated the relative standard deviation of average air Hg values (rel.sd = 0.06) per site (including both years 2015 and 2017) to better represent the variation of Hg(0) among the sites. The relative standard deviation of daily Hg uptake rates of all tree species (Sect. 3.2) equals 0.64. We therefore will amend Section 3.2: "We were not able to normalize daily foliar Hg uptake rates with atmospheric Hg(0) concentrations at each respective sampling site and sample life period, as air Hg(0) measurements were unavailable for our sampling sites. The relative standard deviation of average air Hg(0) concentrations at 6 European measurement sites within the EMEP network between May and Sept. 2015 and 2017 (see Table S2 for details) was 0.06, which is lower than the relative standard deviation of the average daily Hg uptake rates between tree species and forest plots of 0.64** (Error! Reference source not found.**). We therefore argue, that the pronounced differences in median daily foliar Hg uptake rates between tree species cannot be exclusively explained by differences in atmospheric Hg(0) concentrations, but rather suggest a tree physiological cause. However, foliar Hg uptake rates should be normalized to ambient atmospheric Hg(0) concentrations, in particular when comparing foliar Hg observation between the northern and southern Hemisphere or over multi-decadal timescales."**

**Additionally, we will add the following Section to the Supplement:**

**"In order to get a better understanding of the variation in atmospheric Hg(0) in Europe during the growing seasons 2015 and 2017, we obtained air Hg data from the European Monitoring and Evaluation Programme (EMEP). Air Hg measurements for 2015 and 2017 were available at 6 stations (Table S2). Selection of stations was based on availability of measurements in Europe at the relevant time intervals. Available measurements from one station (Iskrba) between May - Sept. 2015 were excluded from the dataset due to abnormally low air Hg values ($0.41 \pm 0.13$ ng m$^{-3}$; mean $\pm$ sd). The temporal frequency of measurements (hourly to 6 days) and consequently the number of measurements varied between the different EMEP stations (Table S2)."**

**Table S 2.** Details on air Hg measurements at 6 EMEP stations during the growing seasons 2015 and 2017.

| Station name | EMEP code | coordinates (lat, lon) | altitude (m) | frequency | time coverage | air Hg (mean $\pm$ sd) (ng m$^{-3}$) | n |
|---|---|---|---|---|---|---|---|
| Andøya | NO0090R | 69.28, 16.01 | 380 | hourly | May - Sept. 2015 | $1.50 \pm 0.09$ | 3371 |
| Auchencorth Moss | GB0048R | 55.79, -3.24 | 260 | 3hourly | May - Sept. 2015 | $1.33 \pm 0.15$ | 1384 |
|  |  |  |  | hourly | May - Sept. 2017 | $1.40 \pm 0.12$ | 2285 |
| BirkenesII | NO0002R | 58.39, 8.25 | 219 | hourly | May - Sept. 2015 | $1.49 \pm 0.24$ | 3402 |
| Diabla Gora | PL0005R | 54.15, 22.07 | 157 | 6 days | May - Sept. 2015 | $1.26 \pm 0.45$ | 23 |
| Iskrba | SI0008R | 45.57, 14.87 | 520 | daily | May - Sept. 2017 | $1.33 \pm 0.80$ | 39 |
| Laheema | EE0009R | 59.5, 25.9 | 32 | hourly | May - July 2015 | $1.40 \pm 0.38$ | 1396 |

**Concerning the possible impact of lower GEM in Southern Europe on the relationship of daily foliar Hg uptake rates with parameters explored in the manuscript, we believe, that this should not be a substantial issue for water VPD, since all VPD sites are located in Switzerland and Germany. We will amend the caption of Fig. 5: "All forest plots are located in Central Europe (latitude 46° - 54°), for which ambient air Hg(0) concentration is relatively constant (see Table S2 and Fig. S6)." With regard to the analysis of daily foliar Hg uptake rates and soil water, the range of latitude of evaluated forest plots is wider (41° - 55°) and potentially the impact of GEM on the presented relationship cannot be ignored. We will mention this caveat at the end of the respective section: "We also suggest determining the possible influence of additional parameters like gravel content and density of soils, tree root depth and atmospheric Hg(0), which could vary within the range of latitude (41° - 55°) of examined forest plots."**

**In Sect. 3.1 we will explicitly include the need to determine GEM when conducting experiments about the sensitivity of foliar Hg uptake to different parameters: "…the sensitivity of species-specific foliar Hg uptake normalized to air Hg(0) concentrations have to be determined in laboratory experiments with regards to elevated VPD, low soil water content or temperature."**

Line 345: Figure 4 not 44

**revised as suggested**

Figure 4: Either use empty space at lower right for figure legend or vertically stack all three panels of the figure.

**For a better overview, we will stack all three panels horizontally, with legend at the bottom.**

Line 414: "In the future"

**revised as suggested**

Line 417: "gravel" is a soil parameter?

**Thank you, correct is gravel content.**

Line 424: When performing 54 linear regressions with a p value of 0.05, you would randomly expect 2 to 3 "significant" ones, even if there aren't any. Therefore, not too much should be made of the findings described on lines 423 to 427.

**After repeating the analysis, we found, that there is no homoscedastic linear regression with a p value below the Bonferroni adjusted p value (here: 0.05/54 = 0.000925), i.e. the significant linear regression coefficients (p < 0.05) could indeed be false positive. We will state so in the manuscript: "None of the resulting 54 sets of linear regression coefficients were significant given a Bonferroni adjusted p-value = 0.000925." We will not further present the findings lines 423 - 427.**

Line 463: "foliage takes up Hg(0) over the entire life time". Should this not be rephrased as "over the entire growing season" as the text earlier admits that mercury uptake during winter is poorly understood?

**Yes, we will rephrase and amend the sentence: "We observed, that foliar Hg concentrations were highly correlated with foliage sampling date (Fig. 2), confirming the notion, that foliage takes up Hg(0) over the entire growing season and over multiple growing seasons in the case of coniferous needles (Fig. 7)."**

Line 464: Again, I think it is necessary to state here that normalisation by the prevailing GEM concentration in the atmosphere is required when comparing foliar Hg uptake rates from different sites. That could be relevant when comparing between foliage from different hemispheres, and between foliage from areas with large differences in mercury source strength.

**We will add the following sentence to the Conclusion: "For reasons of comparability, foliar Hg uptake rates should ideally be normalized to ambient air Hg(0) concentrations when large variation in atmospheric Hg(0) is expected (e.g. between northern and southern hemispheres, in polluted regions or over long timescales).**

---

## Author Comment (AC3)

**Response to Håkan Pleijel**

The referee comments are in black, while the author comments are in **bold print** and blue.

The authors of this paper have analyzed how meteorological/climate variables as well as leaf traits affect the uptake rate of mercury (Hg) of leaves/needles in a range of tree species over a substantial part of Europe. The data base is large, which is an important asset in this type of analysis, formed by observations made using the ICP Forest level II plots and in addition data from a dense sampling network in Austria. Important results include the relationship of foliar Hg uptake rate with leaf nitrogen concentration and leaf mass per area (LMA) as well as links with soil moisture, water vapour pressure deficit (VPD) and other meteorological factors, suggesting a close link of Hg uptake rate with stomatal conductance and physiological activity. In general, this is a well-written and valuable piece of work which should be published. However, improvements are possible, and a number of mostly relatively minor changes and amendments should be considered.

**Thank you for the constructive feedback and your comments.**

Specific comments:

Line 41: "simulated start of the growing season" – some more details about how the simulation was made should be included.

**We agree that the description of the model came off too shortly in the main manuscript. We will therefore add: "To evaluate the beginning of the growing season for deciduous trees, we applied a threshold based growing season model (de Beurs and Henebry, 2010) using the leaf area index (LAI) product by Copernicus Global Land Service based on PROBA-V satellite imagery at a resolution of 300 m and 10 days (Dierckx et al., 2014; Fuster et al., 2020). This growing season model follows an approach by (Bórnez et al., 2020) and defines the beginning of the growing season as the point in time, at which the LAI exceeds the 30% percentile threshold of the amplitude between minimum LAI early in the year and maximum LAI at peak season. For technical details of the model, modelling procedure and quality assurance please refer to supplementary information, Sect. S2.2."**

Line 98: consider removing "balanced", it does not become clear what this refers to in this context, and it is unnecessary to include this word.

**We will delete "balanced".**

Lines 112-118. Information about which needle age classes were harvested should be added as well as which needle age classes were used in different analyses.

**We describe the number of needles samples of different age classes in the last part of this Sect. 2.1: "Coniferous samples consist of needles of different age classes: most of the needle samples (n = 1958) flushed in the sampling season (current season; $y_0$), 600 samples are one-year old ($y_1$), 121 samples are two-year old ($y_2$), 125 are three-year old ($y_3$), 22 samples are four-year old ($y_4$), 60 samples are five-year old ($y_5$) and 3 samples are six-year old ($y_6$) needles." To this, we will add the sentence: "All**

**analysis of this study concerning tree species, foliage structure, nutrient contents and meteorological and site-specific parameters (Sect. 3.1 – 3.6) are based on Hg values of current-season ($y_0$) foliage.”**

Line 117: “typically”? Please be more specific.

**Yes, “typically” is an unnecessary filler word here, so we will delete it.**

Line 173: typo, “factors” should be “factor”.

**revised as suggested**

Line 204 “leaf unfolding” seems not to be the appropriate expression here!? Rather: “emergence of the new flush of needles”? A similar comment applies to the text under 3.1 of the supplementary information. Here it is stated (line 5 and line 22 on page 3) that it is the beginning of the “growing season” that is determined from the PEP725 database. But isn´t it again the emergence of the new flush of needles which is treated here? The growing season of the older needles and thus “the beginning of the growing season of coniferous tree species” (line 5) – with stomatal gas exchange - starts much earlier.

**We agree, that a more specific description of the beginning of the growing season of coniferous trees is necessary to avoid confusion, as there is no single point in time, at which the growing season of coniferous trees starts, which depends on the needle age class. We based the main parts of the analysis of this study on values in current-season needles, which indeed makes the emergence of the new flush of needles the starting point for the analysis. We will change the respective sentences: “While dates of harvest were available for all samples, we determined the start of the growing season of current-season foliage by combining available data sources with start-of-season modelling. These data sources comprise in-situ phenological observations, which were available for 15% of samples, and observations of the emergence of the new flush of needles of coniferous tree species from the Pan European Phenological database PEP725.”**

**In the Supplement we will correct accordingly: “We matched observations on the beginning of the growing season of current-season needles from the Pan European Phenological database PEP725.” Later in the same Section we will explicitly describe the mentioned BBCH codes: “We used data for the beginning of the season (needle age class: $y_0$) of BBCH codes 10, 11, 13, 31, 60, 61 and 223. These BBCH codes correspond to the following growth stages: first leaves separated (BBCH 10); first true leaf, leaf pair or whorl unfolded, first leaves unfolded (BBCH 11); 3 true leaves, leaf pairs or whorls unfolded (BBCH 13); leaf unfolding (>=50%) (BBCH 223); rosette 10% of final length (BBCH 31); first flowers open (BBCH 60); beginning of flowering (BBCH 61).”**

Lines 275-278 and other places: there is a strong focus on current year needles in the study, although not completely. The authors should explain why this is the case. Most conifers retain their needles ~4-10 years and Hg accumulation will continue over several years. In the years after the first, evergreen conifer needles will have a longer period of physiological activity and uptake of Hg, starting earlier in spring and ending later in autumn, compared to broadleaved trees. Thus, even if the uptake rate of Hg per day (mostly used in the paper) is smaller for conifers, this will partly be offset by the longer duration over the year of gas

exchange in needles older than current. This is significant when analyzing biogeochemical fluxes on a (multi)annual scale. To focus only on the uptake rate per day obscures the importance of the variability in the duration of Hg uptake over the year in different types of trees.

**We analyzed the relationship of physiological/climatic parameters with foliage Hg uptake rates, which correspond to Hg accumulation on a leaf/needle basis and do not agree with foliage Hg uptake fluxes (units of Hg per time period over unit ground area). Most needle samples of the dataset were harvested in late fall/early winter, while most broadleaves were sampled during peak season. Normalization for the number of days between spring and sampling day therefore ensures comparability between foliage Hg concentrations of the 23 tree species within the dataset, given that all samples represent current-season foliage. As you rightly point out, daily Hg uptake rates of needles 1 year and older might differ systematically from daily Hg uptake rates of all current-season foliage due to an early start of the growing season and a potential decrease in uptake rates with decreasing physiological activity of older needles. Comparability is the reason for analyzing exclusively current-season needles, with the exception of results presented in Fig. 7, which focuses on Hg concentrations of different needle age classes. We believe it is important to disentangle any confusion arising from the different reference units of Hg concentrations, uptake rates and fluxes. Thus, we suggest to amend Sect. 2.4: "We therefore calculated foliar Hg uptake rates (in ng Hg $g^{-1}_{d.w.}$ $d^{-1}$) of current-season samples by normalizing foliar Hg concentrations to their respective life period in days from the beginning of the growing season (emergence of new foliage) to date of harvest. These resulting foliar Hg uptake rates represent the average daily net Hg accumulation per gram dry weight on a leaf basis and should not be confused with foliar Hg fluxes on a whole-tree basis. Needles of age 1 year or older were excluded from calculating daily foliage Hg uptake fluxes, since Hg uptake might slow down in physiologically less active older needles (Wohlgemuth et al., 2020) and it is unclear, to which extend Hg uptake occurs in older needles in winter and in early spring before the emergence of new foliage."**

**To our knowledge, there is no study describing Hg uptake at the transition of winter to early spring before the emergence of current-season needles. It is an interesting idea to investigate the relevance of the early-spring uptake of needles $y_1 − y_n$ for the overall foliar Hg uptake flux.**

Lines 287-288: as the authors state, the otherwise very informative paper Zhou et al (2021) does not distinguish between different needle age classes, which makes it hard to compare with needle age specific Hg concentration data. Some other reference should be used to support the statement.

**We encountered difficulties to find studies, which compare Hg concentrations in broadleaves and current-season coniferous needles normalized for respective sample life periods, during which foliage was physiologically active. The paper Zhou et al. 2021 has the advantage, that their findings are based on a large dataset. However, given the lack of reporting on needles age classes in Zhou et al. 2021, we will add: "Similarly, Navrátil et al., 2016 reported higher foliar Hg concentrations in deciduous beech leaves (36.3 ng Hg $g^{-1}$) than in coniferous current-season spruce needles (14.1 ng Hg $g^{-1}$) of two adjacent forest plots sampled during peak season (August). Higher Hg concentrations in deciduous leaves (median: 28 ng Hg $g^{-1}$ from 341 remote sites) than in composite multi-age coniferous needles (median: 15 ng Hg $g^{-1}$ from 535 remote sites) were also reported in a global literature compilation (Zhou et al., 2021)."**

Line 338: should be "Leaf Mass per Area".

**revised as suggested**

Line 345: should be "Figure 4" (not 44).

**thank you, corrected**

Figure 4: This figure contains very interesting information, especially the relationship between daily Hg uptake rate and foliar N concentration (representing physiological activity). Such a relationship could become very useful for large-scale modelling of Hg fluxes if supported also by further data. However, it is confusing that the daily Hg uptake rates, especially of the conifers, in Figure 4 are substantially higher than those reported for the same species in Table 1 based on a larger number of observations. Also, the N concentration values differ between Figure 4 and Table 1 for the same species. The authors should discuss these discrepancies, their causes and implications.

**For an overview, we assembled all values in a table, highlighting pine and spruce. Average values of beech, oak, hornbeam and Douglas fir fall within the respective range of ± one sd. to each other.**

| | relatively larger dataset **(Table 1)** | | | relatively smaller sub-dataset **(Fig. 4)** | | |
|---|---|---|---|---|---|---|
| | | | | | | |
| Species group | daily Hg uptake (ng Hg g$^{-1}$$_{d.w.}$ d$^{-1}$) (mean ± sd.) | foliar N conc. (mg N g$^{-1}$d.w.) (mean ± sd.) | n | daily Hg uptake (ng Hg g$^{-1}$$_{d.w.}$ d$^{-1}$) (mean ± sd.) | foliar N conc. (mg N g$^{-1}$d.w.) (mean ± sd.) | n |
| beech | 0.25 ± 0.05 | 23.1 ± 2.9 | 312 | 0.26 ± 0.05 | 23.0 ± 2.8 | 164 |
| oak | 0.20 ± 0.05 | 25.1 ± 2.8 | 252 | 0.22 ± 0.05 | 24.7 ± 2.7 | 106 |
| hornbeam | 0.19 ± 0.03 | 19.4 ± 2.1 | 10 | 0.19 ± 0.04 | 18.9 ± 1.4 | 9 |
| Douglas fir | 0.13 ± 0.02 | 17.0 ± 3.5 | 26 | 0.12 ± 6e-5 | 19.0 ± 2.2 | 2 |
| pine | 0.06 ± 0.02 | 14.4 ± 3.0 | 355 | 0.10 ± 0.02 | 16.1 ± 2.1 | 35 |
| spruce | 0.08 ± 0.02 | 12.9 ± 1.7 | 1509 | 0.11 ± 0.02 | 14.5 ± 1.0 | 33 |

**Median values of daily Hg uptake rates and N concentrations of pine and spruce coincide with respective average values.**

| | relatively larger dataset **(Table 1)** | | relatively smaller sub-dataset **(Fig. 4)** | |
|---|---|---|---|---|
| | | | | |
| | daily Hg uptake (ng Hg g$^{-1}$$_{d.w.}$ d$^{-1}$) median (min – max) | foliar N conc. (mg N g$^{-1}$d.w.) median (min – max) | daily Hg uptake (ng Hg g$^{-1}$$_{d.w.}$ d$^{-1}$) median (min – max) | foliar N conc. (mg N g$^{-1}$d.w.) median (min – max) |
| pine | 0.06 (0.03 – 0.15) | 14.2 (6.2 – 23.3) | 0.10 (0.06 – 0.13) | 16.2 (11.7 – 21.4) |
| spruce | 0.08 (0.03 – 0.17) | 12.8 (1.0 – 25.0) | 0.11 (0.09 – 0.17) | 14.5 (12.3 – 17.0) |

**Density plots reveal that by tendency, the sub-dataset of Fig. 4 is indeed skewed towards higher values of daily Hg uptake rates and N concentrations for pine and spruce compared to the larger dataset of Table 1. This is not the case for beech and oak.**

[Figure]

**Relatively higher/lower average daily foliar Hg uptake rates of pine and spruce are associated with respective higher/lower average N concentrations between the two datasets (compare density plots above). We suggest that this concurrence is consistent with the relationship between Hg and N contents described in Sect. 3.3 and Sect. 3.4. The Hg/N sub-datasets (data Fig. 4) of pine and spruce consist of 10% and 2% of data from the main dataset (data Table 1) respectively, so the shift in average Hg/N values could be coincidental. It is noticeable, however, that the main dataset of Table 1 contains a high proportion of Hg and N measurements from the Austrian Bio-Indicator Grid, while the sub-dataset is exclusively composed of values from ICP Forests Level II Plots. The average daily Hg uptake rate between all sites of the Austrian Bio-Indicator Grid is $0.05 \pm 0.01$ ng Hg g$^{-1}$d.w. d$^{-1}$ (mean $\pm$ sd; n = 162 sites of 2015 and 2017) for pine and $0.08 \pm 0.02$ ng Hg g$^{-1}$d.w. d$^{-1}$ (mean $\pm$ sd; n = 1274 sites of 2015 and 2017) for spruce. The average daily Hg uptake rate between all ICP Forests plots is $0.07 \pm 0.02$ ng Hg g$^{-1}$d.w. d$^{-1}$ (mean $\pm$ sd; n = 80 plots of 2015 and 2017) for pine and $0.09 \pm 0.02$ ng Hg g$^{-1}$d.w. d$^{-1}$ (mean $\pm$ sd; n = 84 plots of 2015 and 2017) for spruce. Most of the pine sampling sites are located in the eastern part of Austria (see Fig. 1 and map below).**

[Figure]

**These Austrian pine sites are characterized by sandy and nutrient-poor soils and pine needles with lower N content compared to e.g. pine samples in Central Germany (see figure below). We therefore assume that the difference of daily needle Hg uptake rates and N concentrations between the datasets presented in Table 1 and Fig. 4 is a result of the geographic shift in nutrient conditions among the sampling sites included in both datasets.**

[Figure]

**We will present all values included in Fig. 4 in a separate table in the Supplement to make these values available. We will also add the density plot to the Supplement to visualize the differences in mean Hg and N between main dataset and sub-dataset.**

Figure 5 (this applies also to figures in the supplementary): While the relationship for pine has a reasonable distribution of data, permitting the authors to derive a quite clear linear relationship, this is not the case for spruce. For the latter species there are two clusters of data separated by a large empty space with respect to the x-variable, which contains no data. Linear regression is not to recommend for such a data set. An option would be to compare and test the difference between the two clusters with a t-test.

**Yes, linear regression for spruce values is not ideal here. A two sided t-test on the two clusters revealed, that the average daily foliar Hg uptake rate is significantly (p = 0.008) different between the two clusters at a water VPD threshold value of 3 kPa. Respective t-tests at all other VPD threshold values revealed non-significant (p > 0.05) differences in average daily Hg uptake rates between the two clusters. For reasons of consistency and comparability with the other datasets, we would prefer to keep presenting results of a linear regression for spruce values in the Supplement, but will amend Sect. 3.5: "Average daily needle Hg uptake rates of spruce needles were clustered between two groups of forest plots with high and low daytime proportions of VPD > threshold (Fig. S9) relative to each other. T-tests revealed a significant (p = 0.008) difference in average daily spruce needle Hg uptake rates between the two clusters for a VPD threshold value of 3 kPa and non-significant (p > 0.05) differences for all other VPD threshold values."**

Line 371: It is not fully explained why the specific water VPD thresholds were used for different species. From where were these thresholds taken?

**The reason for working with water VPD thresholds is, that we were primarily interested in the length of time period of high evaporative demand (time periods of VPD > threshold) within the growing season, during which foliar Hg(0) uptake should be reduced according to theory of stomatal closure. The values of 1.2 kPa, 1.6 kPa, 2 kPa and 3 kPa represent incremental test steps within a VPD range, in which trees show a species-specific stomatal response. We explain this in Sect. 2.5: "We chose these four VPD thresholds because they were reported in literature to incrementally induce leaf stomatal closure of temperate forest trees, ranging from initial stomatal closure (around 0.8 kPa – 1 kPa (Körner, 2013)) to maximum stomatal closure (at around 3 kPa – 3.2 kPa (CLRTAP, 2017))." To make it clear, that these threshold values were chosen as test values we will add: "We chose these four VPD thresholds as test values because they were reported…"**

Line 389: replace "had" by "has".

**revised as suggested**

Lines 393-394: "… the high degree of stomatal closure under drought stress of spruce". I do not understand this statement in relation to the data presented. The average daily Hg uptake rate, suggested by authors to be a possible proxy for stomatal conductance, does not differ very much between high and low VPD (Figure 5b), indicating a small response in stomatal conductance by high water VPD in spruce!?

**Both spruce and pine respond to rising water VPD by reducing their stomatal conductance (Zweifel et al., 2009). Regarding daily foliar Hg uptake rates, we detected a significant linear regression coefficient versus a time proportion above a threshold of VPD > 3 kPa for spruce and a threshold of 1.2 kPa for pine (compare threshold values of Fig. 5a and 5b). From literature we derive, that spruce is a bit slower to respond to rising VPD as compared to pine, but that spruce stomata are almost completely closed at high VPD like 3 kPa (Zweifel et al., 2009). We will refine the description of findings from literature: "Among isohydric species, pine has been discovered to reduce tree conductance and stomatal aperture during the onset of dry conditions earlier and at a greater rate than spruce (Lagergren and Lindroth, 2002; Zweifel et al., 2009). Spruce was observed to keep stomata almost completely closed under drought stress, i.e. high VPD and/or soil water deficit (Zweifel et al., 2009)."**

Figure 7: Although there is a general trend for higher Hg concentrations in older needles in Figure 7, also seen in other studies, the pattern in the figure is not completely clear. This may partly be caused by the heterogeneity of the data, with a strongly varying number of observations for the different needle age classes. It may also be the case that the larger number of needle age classes included in the figure compared to most other studies, indicate a levelling off in the rate of annual increase in Hg concentration of the oldest needles. The authors should discuss the data presented in Figure 7 in further depth including the possible consequences of the heterogeneity of the data. The quite strong levelling off in annual Hg concentration increase for y4-y6 in Figure 7 is not in complete agreement with the statement on lines 432-436.

**We will discuss in more detail: "In agreement with previous studies (Ollerova et al., 2010; Hutnik et al., 2014; Navrátil et al., 2019; Wohlgemuth et al., 2020; Pleijel et al., 2021), we found a trend of Hg concentrations in differently aged needles with older needles exhibiting higher Hg concentrations (Fig. 7), demonstrating that Hg accumulation continues in older needles. Annual Hg net accumulation seems to slow down in older spruce needles of age classes $y_3 - y_6$ in contrast to needles of age classes $y_0 - y_2$ (Fig. 7), albeit ranges of average Hg concentrations ± standard deviation overlap among older and younger spruce needles, which might be the result of relatively lower sample numbers of older needles compared to younger needles (e.g. 3 samples for $y_6$ vs. 301 samples for $y_1$). A decline in foliar Hg uptake by older needles could be caused by lower physiological activity, cuticular wax degradation or an increase in Hg re-emission with needle age (Wohlgemuth et al., 2020)."**

Lines 450-453 on stomatal conductance modelling: the phrase "model in a multiplicative way" is not appropriate and hard to understand for people not strongly involved in stomatal conductance modelling. Also, there exist other types (e.g., photosynthesis based) stomatal conductance models than multiplicative which could be considered. It would be appropriate to use a more recent reference than Emberson et al (2000), since a lot of development has taken place in stomatal conductance modelling over the last two decades. The authors may consider Emberson et al (2018) Ozone effects on crops and consideration in crop models. *European Journal of Agronomy* 100, 19–34, although it is on crops.

**Thank you, we will delete "in a multiplicative way" and add the suggested reference.**

Line 480-481: The suggestion by the authors to assume that foliar Hg can represent the stomatal conductance (integrated over longer time scales) is interesting and thought-provoking. This statement must be based on the assumption that the atmospheric concentration of Hg is essentially constant, from year to year and from place to place. To what extent is this assumption valid? This should be discussed.

**Based on comments by reviewer Frank Wania, we will add a discussion on GEM to Sect. 3.2 and will comment on the topic throughout the manuscript. Generally, the horizontal variability of GEM during the season is relatively low. We evaluated a GEM concentration of 1.39 ± 0.09 ng m$^{-3}$ (mean ± sd) based on average GEM concentrations at 6 EMEP stations in Central Europe between May – Sept. In principle, however, we agree that atmospheric GEM should be taken into account and revised the text in the manuscript accordingly: "We therefore suggest, that foliar Hg measurements bear the potential to serve as a proxy for stomatal conductance, providing a time-integrated measure for stomatal aperture when taking into account the spatial and temporal variation in atmospheric Hg(0)."**

In the Conclusion part it would be appropriate to discuss the relationship between the net accumulation in leaves/needles over time periods of weeks to years, which is the focus in the paper, and the dynamic bi-directional fluxes of Hg to/from vegetation observed in many studies using highly time resolved measurements.

**We are not sure, if a discussion about short-scale temporal (uptake/re-emission) fluxes in the Conclusion will advance the main message of this manuscript as the focus of this study is on long-term foliar net Hg accumulation rates. However, we will add an explanatory sentence to Sect. 2.4:**

**"Please also note, that daily foliar Hg uptake rates in this study represent average values over the growing season. The actual daily foliar Hg uptake on a given day might differ from the average value depending on the time period within the growing season (e.g. early season vs. peak season) (Laacouri et al., 2013)."**

Essentially, this is a very interesting and valuable paper.

**Thank you very much for this kudo.**